# FBN-1, a fibrillin-related protein, is required for resistance of the epidermis to mechanical deformation during *C. elegans* embryogenesis

Melissa Kelley[1†], John Yochem[1†], Michael Krieg[2,3†], Andrea Calixto[4,5], Maxwell G Heiman[6,7], Aleksandra Kuzmanov[1], Vijaykumar Meli[8], Martin Chalfie[4], Miriam B Goodman[2], Shai Shaham[9], Alison Frand[8], David S Fay[1*]

[1]Department of Molecular Biology, University of Wyoming, Laramie, United States; [2]Department of Molecular and Cellular Physiology, Stanford University, Stanford, United States; [3]Department of Chemical Engineering, Stanford University, Stanford, United States; [4]Department of Biological Sciences, Columbia University, New York, United States; [5]Center for Genomic and Bioinformatics, Universidad Mayor, Santiago, Chile; [6]Department of Genetics, Harvard Medical School, Boston Children's Hospital, Boston, United States; [7]Division of Genetics, Boston Children's Hospital, Boston, United States; [8]Department of Biological Chemistry, David Geffen School of Medicine, University of California, Los Angeles, United States; [9]Laboratory of Developmental Genetics, The Rockefeller University, New York, United States

**Abstract** During development, biomechanical forces contour the body and provide shape to internal organs. Using genetic and molecular approaches in combination with a FRET-based tension sensor, we characterized a pulling force exerted by the elongating pharynx (foregut) on the anterior epidermis during *C. elegans* embryogenesis. Resistance of the epidermis to this force and to actomyosin-based circumferential constricting forces is mediated by FBN-1, a ZP domain protein related to vertebrate fibrillins. *fbn-1* was required specifically within the epidermis and FBN-1 was expressed in epidermal cells and secreted to the apical surface as a putative component of the embryonic sheath. Tiling array studies indicated that *fbn-1* mRNA processing requires the conserved alternative splicing factor MEC-8/RBPMS. The conserved SYM-3/FAM102A and SYM-4/WDR44 proteins, which are linked to protein trafficking, function as additional components of this network. Our studies demonstrate the importance of the apical extracellular matrix in preventing mechanical deformation of the epidermis during development.

*For correspondence: davidfay@uwyo.edu

†These authors contributed equally to this work

Competing interests: The authors declare that no competing interests exist.

## Introduction

In addition to their essential protective, structural and physiological functions, epithelial cells and their closely associated extracellular matrices (ECMs) serve as important mediators of embryonic morphogenesis and organogenesis (*Davidson, 2011*, *2012*; *Heisenberg and Bellaiche, 2013*). These developmental functions require epithelial tissues to be appropriately resistant to deformation by a variety of intrinsic and extrinsic mechanical forces that arise during the normal course of development. Accordingly, an improper force balance can lead to morphological abnormalities and birth defects (*Epstein et al., 2004*; *Moore et al., 2013*).

In *Caenorhabditis elegans*, the outermost epithelial layer or epidermis (commonly called the hypodermis in nematodes) is initially established during early-to-mid embryogenesis

**eLife digest** For an animal embryo to develop, its cells must organize themselves into tissues and organs. For example, skin and the lining of internal organs—such as the lungs and gut—are made from cells called epithelial cells, which are tightly linked to form flat sheets.

In a microscopic worm called *Caenorhabditis elegans*, the outermost layer of epithelial cells (called the epidermis) forms over the surface of the embryo early on in embryonic development. Shortly afterwards, the embryonic epidermis experiences powerful contractions along the surface of the embryo. The force generated by these contractions converts the embryo from an oval shape to a roughly cylindrical form. These contractions also squeeze the internal tissues and organs, which correspondingly elongate along with the epidermis.

It has been known for decades that such 'mechanical' forces are important for the normal development of embryos. However, it remains poorly understood how these forces generate tissues and organs of the proper shape—partly because it is difficult to measure forces in living embryos. It is also not clear how the mechanical properties of specific tissues are controlled.

Now, Kelley, Yochem, Krieg et al. have analyzed the development of *C. elegans*' embryos and discovered a novel mechanical interplay between the feeding organ (called the pharynx) and the worm's epidermis. The experiments involved studying several mutant worms that perturb epidermal contractions and disrupt the attachment of the pharynx to the epidermis. These studies suggested that the pharynx exerts a strong inward pulling force on the epidermis during development. Using recently developed methods, Kelley, Yochem, Krieg et al. then measured mechanical forces within intact worm embryos and demonstrated that greater forces were experienced in cells that were being pulled by the pharynx.

Kelley, Yochem, Krieg et al. further analyzed how the epidermis normally resists this pulling force from the pharynx and implicated a protein called FBN-1. This worm protein is structurally related to a human protein that is affected in people with a disorder called Marfan Syndrome. Worm embryos without the FBN-1 protein become severely deformed because they are unable to withstand mechanical forces at the epidermis. FBN-1 is normally synthesized and then transported to the outside of the worm embryo by epidermal cells, where it is thought to assemble into a meshwork of long fibers. This provides a strong scaffold that attaches to the epidermis to prevent the epidermis from undergoing excessive deformation while it experiences mechanical forces.

The work of Kelley, Yochem, Krieg et al. provides an opportunity to understand how FBN-1 and other fiber-forming proteins are produced and transported to the cell surface. Moreover, these findings may have implications for human diseases and birth defects that result from an inability of tissues to respond appropriately to mechanical forces.

(∼400 min post fertilization; *Sulston et al., 1983*). At this time, future epidermal cells execute stereotypical movements, shape changes and migrations to produce a 1.5-fold-stage embryo that is surrounded by an epithelium consisting of a single cell layer (*Sulston et al., 1983*; *Chisholm and Hardin, 2005*; *Chisholm and Hsiao, 2012*). Shortly after this stage, ring-shaped actomyosin bundles, which are spaced regularly along the anteroposterior axis of the embryo, undergo coordinated contraction. This contraction leads to the circumferential constriction of the embryo and its conversion to a tapered cylindrical (fusiform) shape that is ∼250 μm long (about five times the length of the egg shell; *Costa et al., 1997*; *Priess and Hirsh, 1986*). As a consequence of constriction at the epidermal surface and contractions by body wall muscles (*Williams and Waterston, 1994*; *Chisholm and Hardin, 2005*), tissues and organs inside the embryo are thought to experience squeezing forces and to elongate in conjunction with the outer layers of the embryo. Notably, the apical ECM (aECM) of the embryonic epidermis, termed the embryonic sheath, is required to prevent excessive constriction and deformation of the epidermis by actomyosin ring contraction (*Priess and Hirsh, 1986*). Although critical for development, the molecular composition and related physical properties of the embryonic sheath remain poorly characterized.

Despite a growing interest in mechanical aspects of development and morphogenesis (*Guillot and Lecuit, 2013*; *Heisenberg and Bellaiche, 2013*), the interplay between mechanical forces and the physical properties and structure of tissues have been difficult to characterize. This is due in part to an

incomplete description of mechanical forces in living embryos. In addition, genetic redundancy has likely impeded progress toward fully understanding the molecular control of tissue and organismal morphogenesis (*Thomas, 1993*; *Pickett and Meeks-Wagner, 1995*; *Tautz, 2000*; *Herman and Yochem, 2005*; *Bussey et al., 2006*).

Here we describe a morphological defect that results from the failure of the anterior epidermis to maintain its proper shape while experiencing an inward-directed pulling force exerted by the developing pharynx (foregut) as it undergoes elongation. This defect occurs at a low frequency in single mutants of *mec-8*, *sym-3* and *sym-4*, but at a high frequency in *mec-8*; *sym-3* and *mec-8*; *sym-4* double mutants, indicating that this process is redundantly controlled (*Davies et al., 1999*; *Yochem et al., 2004*). Whereas *sym-3* and *sym-4* encode conserved proteins with predicted roles in vesicular trafficking (*Yochem et al., 2004*; also see 'Discussion'), *mec-8* encodes a conserved RNA-binding protein involved in alternative splicing (*Lundquist et al., 1996*; *Spike et al., 2002*). We have shown that the contribution of MEC-8 in the resistance to this force arises, at least in part, through its control of FBN-1, a protein that shares several domains with vertebrate fibrillins and acts in the embryonic sheath. Notably, mutations in human fibrillin genes lead to connective tissue disorders including Marfan syndrome (*Dietz et al., 2005*; *Ramirez and Dietz, 2009*; *Ramirez and Sakai, 2010*).

## Results

### Morphological defects in *mec-8*; *sym-3* and *mec-8*; *sym-4* mutants are caused by an inward-directed pulling force exerted by the pharynx on the epidermis

In wild-type embryos at the 1.5-fold stage of development, a shallow pit (~2.1 µm deep), termed the sensory depression, is detected in the region corresponding to the location of the future mouth (buccal cavity; *Figure 1A*, *Table 1*; *Sulston et al., 1983*). This morphological feature is relatively short-lived and is no longer visible in threefold-stage embryos (*Figure 1A*, *Figure 2C*). In contrast, *mec-8*; *sym-3* and *mec-8*; *sym-4* embryos had a striking keyhole-shaped invagination in this region, which increased in depth between the 1.5-fold (~4.3 µm) and 3-fold (~9.5 µm) stages (*Figure 1A*, *Table 1*). In contrast to wild-type L1 larvae, in which the pharynx and associated buccal capsule (terminal mouth part) extended to the anterior tip of the worm, *mec-8*; *sym-3* and *mec-8*; *sym-4* L1 larvae displayed what we have termed the 'Pharynx ingressed' (Pin) phenotype, in which the pharynx and buccal capsule are displaced toward the posterior end of the animal (*Figure 1A*). In Pin larvae, lateral anterior tissues appeared to fold over and surround the ingressed buccal capsule, thereby preventing double mutants from feeding (*Figure 1A*). Although these defects were observed at only low frequencies in *sym-3*, *sym-4* and *mec-8* single mutants, they were highly penetrant in *mec-8*; *sym-3* and *mec-8*; *sym-4* double mutants (*Figure 1B*, *Supplementary file 1*).

To account for the defects observed in *mec-8*; *sym-3* and *mec-8*; *sym-4* double mutants, we proposed a testable model for pharyngeal and embryonic elongation. As described above, the embryo acquires an elongated shape through the circumferential constriction of ring-shaped actomyosin bundles arrayed along the surface of the epidermis (*Priess and Hirsh, 1986*). During initial stages of embryonic morphogenesis (~350–380 min), the primordial pharynx exists as a ball of cells with no connection to the future mouth (buccal capsule) or epidermis (*Figure 1C*). Linkage of the pharynx to the mouth and epidermis is established between the comma and 1.5-fold stages (~380–410 min; *Figure 1C*, data not shown; *Sulston et al., 1983*; *Portereiko and Mango, 2001*). During embryonic development, the pharynx lengthens along its anteroposterior axis, transforming from a blunt conical shape into a bi-lobed structure that is attached to the mouth at the anterior and to the intestine in the mid body (*Figure 1C*). We hypothesized that lengthening of the pharynx is facilitated in part by an outward-directed pulling force that is exerted by the anterior epidermis as the embryo undergoes elongation. In addition, as the pharynx is stretched, it exerts a counter inward-pulling force on the embryonic epidermis. This inward-pulling force would be greatest in the region where the pharynx attaches to the epidermis, contributing to the formation of the sensory depression (*Figure 1C*). We liken this situation to that of a spring that is attached (on the inside) to the 'anterior' end of an elastic-walled cylinder, with the cylinder representing the embryonic epidermis and the spring representing the pharynx (*Figure 1C*). The 'posterior' end of the spring in this model is held in place within the middle of the cylinder through localized contacts, which in the case of the pharynx most likely occur through cell–cell interactions. As the cylinder elongates, it stretches the spring, which

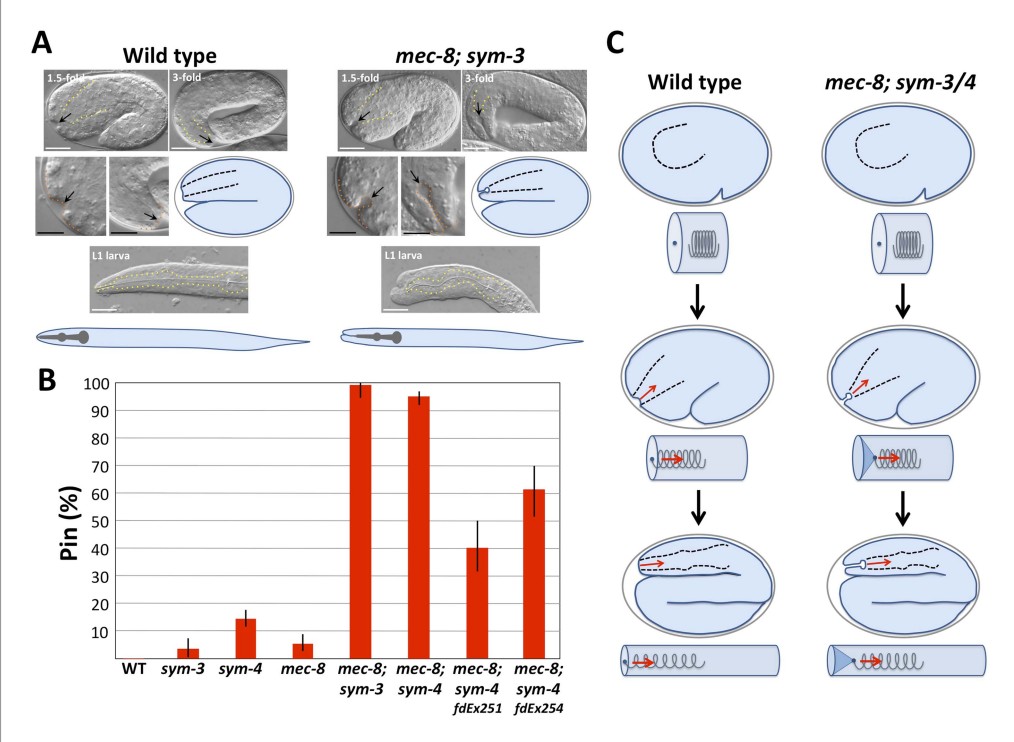

**Figure 1**. *mec-8*; *sym-3* and *mec-8*; *sym-4* mutants exhibit an abnormal ingression of the anterior epidermis. (**A**) Whereas wild-type 1.5-fold embryos display only a shallow ingression of the anterior epidermis (sensory depression) and little or no ingression by the threefold stage, *mec-8*; *sym-3* and *mec-8*; *sym-4* (data not shown) mutants contain a deep keyhole-shaped ingression that increases in depth between the 1.5-fold and 3-fold stages. *mec-8*; *sym-3* and *mec-8*; *sym-4* (data not shown) L1 larvae also contain an ingressed pharynx (Pin) and associated deformities in the head region. Yellow dashed lines indicate lateral pharyngeal borders; orange dashed lines, the sensory depression or keyhole; black arrows, posterior extent of ingression. White scale bars = 10 µm, black bars = 5 µm. (**B**) Quantification of the Pin phenotype in single and double mutants and in *mec-8*; *sym-4* double mutants containing multi-copy extrachromosomal arrays (*fdEx251* and *fdEx254*) that express the *fbn-1e* cDNA isoform under the control of the native *fbn-1* promoter. Error bars represent 95% CIs. For additional details, see *Table 1* and *Supplementary file 1*. (**C**) Spring-and-cylinder model in which the pharynx exerts an inward-pulling force at the anterior epidermis throughout the mid-to-late stages of embryonic morphogenesis. In embryo representations, pharyngeal borders are indicated by black dashed lines; in cylindrical representations, the pharynx is represented by a spring that is attached to the anterior epidermis at the dark blue dot. Early comma, 1.5-fold and 3-fold stages of embryogenesis are depicted. Red arrows indicate the inward-pulling force on the epidermis that results from the resistance of the pharynx to stretching.

then exerts an inward-pulling force at the site of attachment to the cylinder wall (*Figure 1C*). We hypothesize that in wild-type embryos, one or more means of structural reinforcement prevents the anterior epidermis from undergoing a pronounced invagination or ingression in response to the pharyngeal pulling force. In contrast, the epidermis in *mec-8*; *sym-3* and *mec-8*; *sym-4* mutants is insufficiently reinforced, due to the combined defects in processes controlled by *mec-8* and *sym-3/4*, resulting in mechanical deformation of the epidermis, the genesis of the keyhole and, ultimately, the Pin phenotype.

One prediction of our model is that prevention of a pharyngeal-epidermal attachment should suppress keyhole formation in *mec-8*; *sym-3/4* embryos (*Figure 2A*). To test this, we used a deletion mutation (*tm3671*) in *pha-1*, which encodes a cytoplasmic protein of unknown function, that prevents initial attachment of the pharynx to the epidermis in ~85% of embryos (*Fay et al., 2004*, *2012*; *Kuzmanov et al., 2014*). As predicted, formation of the keyhole was suppressed in *mec-8*; *pha-1* (*tm3671*); *sym-3* triple mutants in which the pharynx failed to attach (*Figure 2A*, *Table 1*). In contrast, in *mec-8*; *pha-1*(*tm3671*); *sym-3* embryos in which the pharynx was attached to the epidermis (~15%),

**Table 1.** Ingression depths of the anterior epidermis

| | Ingression depth (µm) ± 95% CI (range; n) | |
| --- | --- | --- |
| Genotype | 1.5-fold | 3.0-fold |
| N2 | 2.12 ± 0.23 (1.09–3.12; 20) | 0.26 ± 0.096 (0.0–0.67; 20) |
| sym-3(mn618) | 2.39 ± 0.40 (0.81–4.24; 24) | 0.48 ± 0.56 (0.0–6.55; 22) |
| sym-4(mn619) | 2.74 ± 0.64 (0.72–5.67; 21) | 1.28 ± 1.06 (0.0–6.74; 22) |
| mec-8(u74) | 2.33 ± 0.46 (0.72–4.34; 20) | 2.42 ± 1.50 (0.0–10.43; 26) |
| mec-8; sym-3* | 4.25 ± 0.89 (2.77–5.72; 18) | 9.82 ± 0.68 (7.84–12.00; 15) |
| mec-8; sym-4 | 4.27 ± 1.16 (2.09–6.45; 16) | 9.19 ± 0.83 (7.07–10.14; 12) |
| Pha-1(tm3671) | 0.87 ± 0.18 (0.45–1.18; 16) | NA |
| mec-8; pha-1(tm3671); sym-3* | 0.83 ± 0.11 (0.40–1.19; 17) | NA |
| pha-1(e2123) | 2.15 ± 0.27 (1.04–3.34; 19) | 0.10 ± 0.07 (0.0–0.59; 16) |
| mec-8; pha-1(e2123); sym-3* | 5.27 ± 0.53 (3.89–7.46; 14) | 0.60 ± 2.35 (0.0–10.29; 19) |
| fbn-1(ns67) | 3.18 ± 0.85 (0.60–6.02; 13) | 5.34 ± 1.31 (0.0–9.24; 20) |
| fbn-1(ns67); sym-3 | 5.20 ± 0.41 (3.82–6.71; 20) | 11.73 ± 0.85 (8.59–16.34; 19) |
| fbn-1(ns67); sym-4 | 5.98 ± 0.55 (4.25–7.66; 12) | 12.84 ± 0.78 (9.65–16.37; 27) |
| mec-8; fbn-1(ns67) | 5.03 ± 0.47 (3.76–7.19; 19) | 9.84 ± 0.55 (6.94–12.24; 21) |
| fbn-1(tm290) | 6.25 ± 1.81 (0.99–12.17; 16) | 7.63 ± 3.66 (0.0–24.17; 17) |
| fbn-1(tm290); sym-3 | 5.65 ± 0.61 (2.59–7.29; 19) | 15.05 ± 1.56 (9.12–26.06; 24) |
| fbn-1(tm290); sym-4 | 5.54 ± 0.86 (3.52–9.12; 17) | 13.10 ± 1.72 (7.22–19.47; 20) |
| mec-8; fbn-1(tm290) | 9.84 ± 0.55 (5.47–15.18; 31) | NA |

*Because these strains give rise to a high frequency of viable mnEx169(–) progeny in the first generation following loss of the array (F1 escapers), next-generation progeny (F2) from mnEx169(–) F1 parents were scored. NA, Non-Applicable; these genotypes led to embryonic arrest prior to the 3-fold stage.

a keyhole was observed, indicating that the loss of attachment per se, rather than the loss of *pha-1* activity, was responsible for the suppression of keyhole formation in the majority of triple mutants (**Figure 2A**).

A second prediction of our model is that maintenance of a pharyngeal-epidermal attachment would be required for persistence of a keyhole in embryos and for progression to a Pin phenotype in larvae (**Figure 2B**). To test this prediction, we used a hypomorphic allele of *pha-1* (*e2123*), which establishes a transient connection between the pharynx and epidermis that is severed at later stages of embryogenesis (**Schnabel and Schnabel, 1990**; **Fay et al., 2004**; **Kuzmanov et al., 2014**). In our cylinder-and-spring analogy, loss of the pharyngeal-epidermal attachment in *pha-1(e2123)* mutants would be akin to severing the spring near the site of attachment to the tube, leading to the ingressed elastic cylinder tip popping back out and recoil of the spring (**Figure 2B**). As predicted by our model, early-stage *mec-8; pha-1(e2123); sym-3* triple mutants formed a stereotypical keyhole, consistent with the presence of a pharyngeal-epidermal connection. The absence of a keyhole or Pin phenotype in late-stage embryos and L1 larvae, however, indicated that anterior ingression of the epidermis requires a sustained pulling force exerted by the pharynx and that the keyholes are not static once formed (**Figure 2B**).

We also observed that the depth of the keyhole in *mec-8; sym-3* and *mec-8; sym-4* mutants steadily increased from the comma stage to the threefold stage of embryogenesis (**Figure 1A**, **Figure 2C**, **Table 1**). We hypothesized that failure to elongate past the 1.5-fold or 2.0-fold stages would, however, prevent further deepening of the keyhole. In our model, this would be akin to lengthening the cylinder only partway, thereby preventing further ingression of the tip. To test this, we inhibited morphogenesis past the twofold stage in *mec-8; sym-4* mutants by RNA interference (RNAi) of *let-502/ROCK*, which encodes an epidermal-expressed Rho-binding kinase required for embryonic elongation (**Wissmann et al., 1997**, **1999**). As expected, keyhole depth in *mec-8; sym-4; let-502* (*RNAi*) embryos increased until the twofold stage, reaching an average depth of ~6 µm, identical to

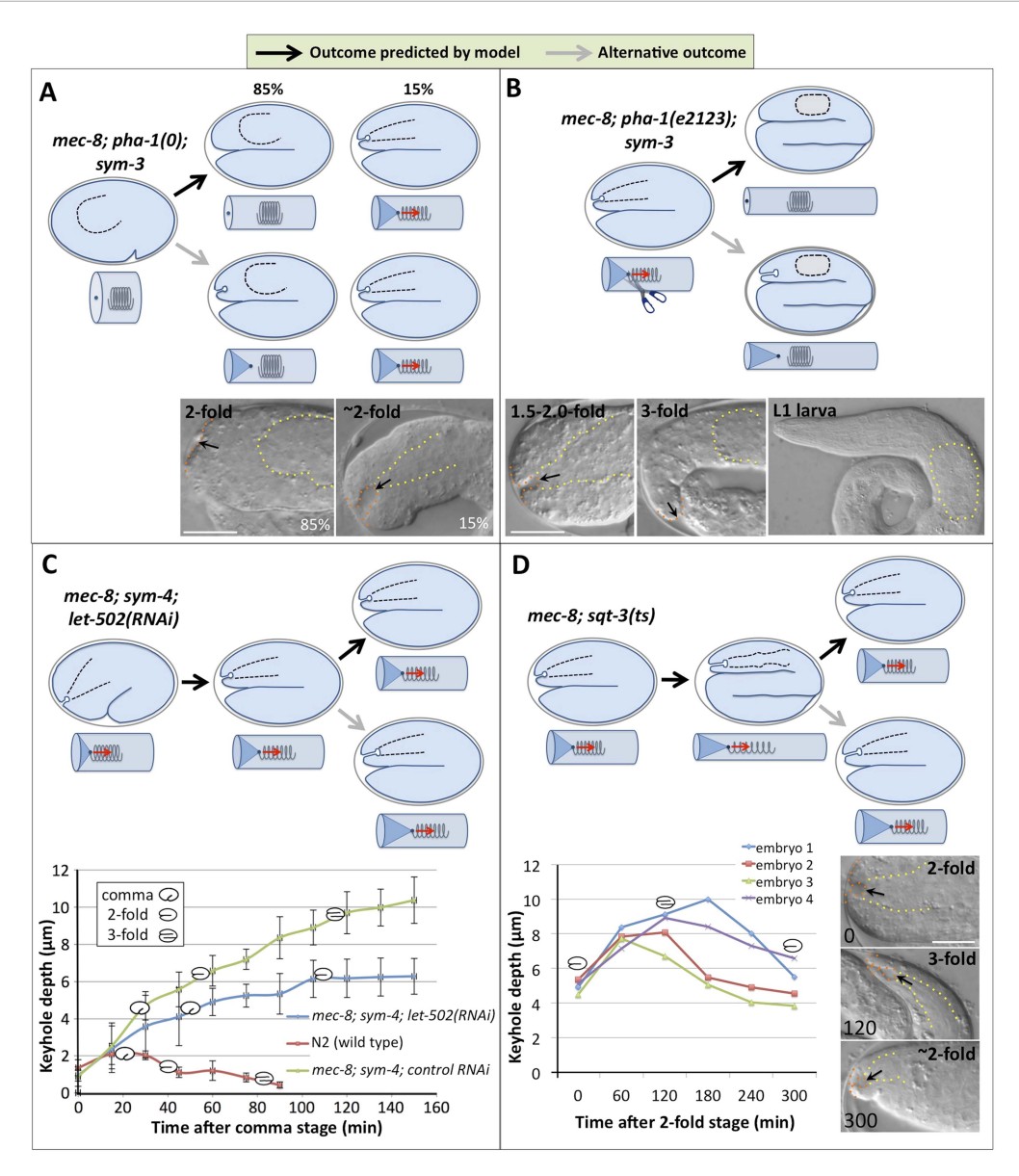

**Figure 2**. Genetic and phenotypic analyses support an extension spring model for pharyngeal elongation. (**A–D**) Predicted models and outcomes for testing the hypothesis that the elongating pharynx exerts an inward-pulling force on the anterior epidermis. Black arrows in models show the predicted (and observed) outcomes; gray arrows, alternative outcomes. For panels with DIC images, yellow dashed lines indicate lateral pharyngeal borders; orange dashed lines, the sensory depression or keyhole; black arrows, posterior extent of ingression. White scale bars = 10 μm. For additional details, see *Table 1* and *Supplementary file 1*. (**A**) In *mec-8*; *pha-1(tm3671)*; *sym-3* mutants that fail to establish a connection between the pharynx and epidermis (85%), deep ingressions or keyholes are not observed, whereas mutants that form an initial attachment (15%) form a stereotypical keyhole. (**B**) Detachment of the pharynx from the epidermis after the twofold stage in *mec-8*; *pha-1(e2123)*; *sym-3* mutants leads to loss of the anterior ingression by the threefold embryonic stage and suppression of Pin in L1 larvae. (**C**) Whereas the depth of the keyhole in *mec-8*; *sym-4* mutants steadily increases between the 2-fold and 3-fold stages of embryogenesis, inhibition of embryonic elongation past the twofold stage by *let-502(RNAi)* prevents further deepening of the ingression. Error bars indicate 95% CIs, and diagrammed embryos denote the approximate stages of development for each genotype; n = 5 for each genotype at each time point. (**D**) Reversal of embryonic elongation in *mec-8*; *sqt-3 (e2117ts)* mutants leads to a decrease in keyhole depth. Each line in the plot represents a different embryo; diagrammed embryos denote the approximate stages of development. For these experiments, rare *mec-8*; *sqt-3(ts)* mutants that exhibited a keyhole at the twofold stage (~5%) were analyzed for reasons of experimental convenience. DOI: 10.7554/eLife.06565.005

that observed for control RNAi-treated *mec-8*; *sym-4* mutants at the same stage of development (*Figure 2C*). After morphogenetic arrest, however, keyholes in *mec-8*; *sym-4*; *let-502(RNAi)* embryos failed to deepen, indicating that the progressive increase in keyhole depth is a function of embryonic elongation, rather than the passage of time. We also observed that *mec-8*; *sym-4*; *let-502(RNAi)* embryos took longer to transit from the comma stage to the twofold stage than the control RNAi-treated strain. Consistent with our model, the rate at which keyhole depth increased in *mec-8*; *sym-4*; *let-502(RNAi)* embryos was reduced in proportion with the delay in embryonic elongation (*Figure 2C*).

A final prediction of our model is that a reversal of embryonic elongation should lead to a consequent reduction in the depth of the keyhole in embryos. In our model, this would be analogous to shortening the cylinder and observing a reduction in anterior tip ingression (*Figure 2D*). To test this, we used a conditional allele of *sqt-3* (*e2117ts*), which undergoes a reversal of elongation from the ~threefold–~twofold stages after temperature upshift (*Kusch and Edgar, 1986*; *Priess and Hirsh, 1986*). Consistent with our model, keyholes reached a maximum depth of ~8–10 μm at around the threefold stage but then shrunk to ~4–6 μm after a partial reversal of embryonic elongation (*Figure 2D*). Taken together, our findings provide strong evidence that resistance of the pharynx to stretching or lengthening leads to an inward-pulling force on the anterior epidermis during much of embryogenesis. In the case of wild-type embryos, this force is resisted to an appropriate extent, and a normal morphology is achieved. In contrast, morphological defects in *mec-8*; *sym-3* and *mec-8*; *sym-4* embryos and larvae suggest that the mechanical properties of the epidermis may be compromised in these mutants, leading to the Pin phenotype.

## A FRET-based tension sensor reveals mechanical forces operating during embryogenesis

To visualize biomechanical forces operating during embryogenesis, we made use of recently developed FRET-based methods for detecting mechanical tension in live cells (*Meng et al., 2008*; *Grashoff et al., 2010*; *Meng et al., 2011*). Specifically, we used strains expressing a tension sensor module (TSMod) inserted into the coding sequence of the *unc-70* gene (*Figure 3A*; *Krieg et al., 2014*). UNC-70, a β-spectrin ortholog, is expressed widely during embryogenesis and acts together with α-spectrin and actin to form a subcortical cytoskeletal network that is critical for cell shape and mechanics in a variety of cell types in *C. elegans* (*Bretscher, 1991*; *Hammarlund et al., 2000*; *Moorthy et al., 2000*; *Norman and Moerman, 2002*). Importantly, the UNC-70(TSMod) fusion protein localized to the cell membrane cortex in a pattern that was seemingly identical to immunostaining of endogenous UNC-70 (*Moorthy et al., 2000*), with a prominent accumulation at future location of the buccal cavity (*Figure 3B*). Moreover, UNC-70(TSMod) rescued the severely paralyzed locomotion phenotype of *unc-70* null mutant animals, indicating that the fusion protein is functional (*Krieg et al., 2014*).

The TSMod sensor consists of a donor (mTFP) and acceptor (Venus) fluorophore separated by a flexible linker made of 40 residues from the spider-silk flagelliform, which acts as an entropic nanospring suitable for estimating biologically relevant forces (*Figure 3A*; *Grashoff et al., 2010*). The linker is sensitive to molecular forces in various systems (*Borghi et al., 2012*; *Morimatsu et al., 2013*; *Cai et al., 2014*; *Krieg et al., 2014*; *Paszek et al., 2014*). Thus, as stretching forces act on this spring the two FRET fluorophores will be pulled apart and lead to a visible change in energy transfer. Consequently, a low FRET index indicates the application of a stretching force to UNC-70 (TSMod) and suggests that actin-spectrin networks in such regions experience high levels of mechanical tension. Conversely, a high FRET index suggests that such regions experience low or no tension across the actin-spectrin network. Importantly, we previously used this same sensor to investigate mechanical tension in *C. elegans* neurons and extend this robust imaging procedure (see 'Materials and methods') to characterize its performance in living animals (*Krieg et al., 2014*).

To quantify the extent to which pharyngeal attachment and subsequent pulling forces lead to higher tension near the sensory depression, we compared FRET at the sensory depression region (SDR) with areas outside the sensory depression (non-SDR) in embryos before and after pharyngeal attachment to the epidermis (early comma and 1.5-fold stages, respectively; see 'Materials and methods'). This strategy allows us to compare pixels from the SDR and non-SDR that have been measured under exactly the same conditions in a pairwise manner, since both measurements were derived from the same image and analyzed identically. Thus, any changes in FRET efficiency are

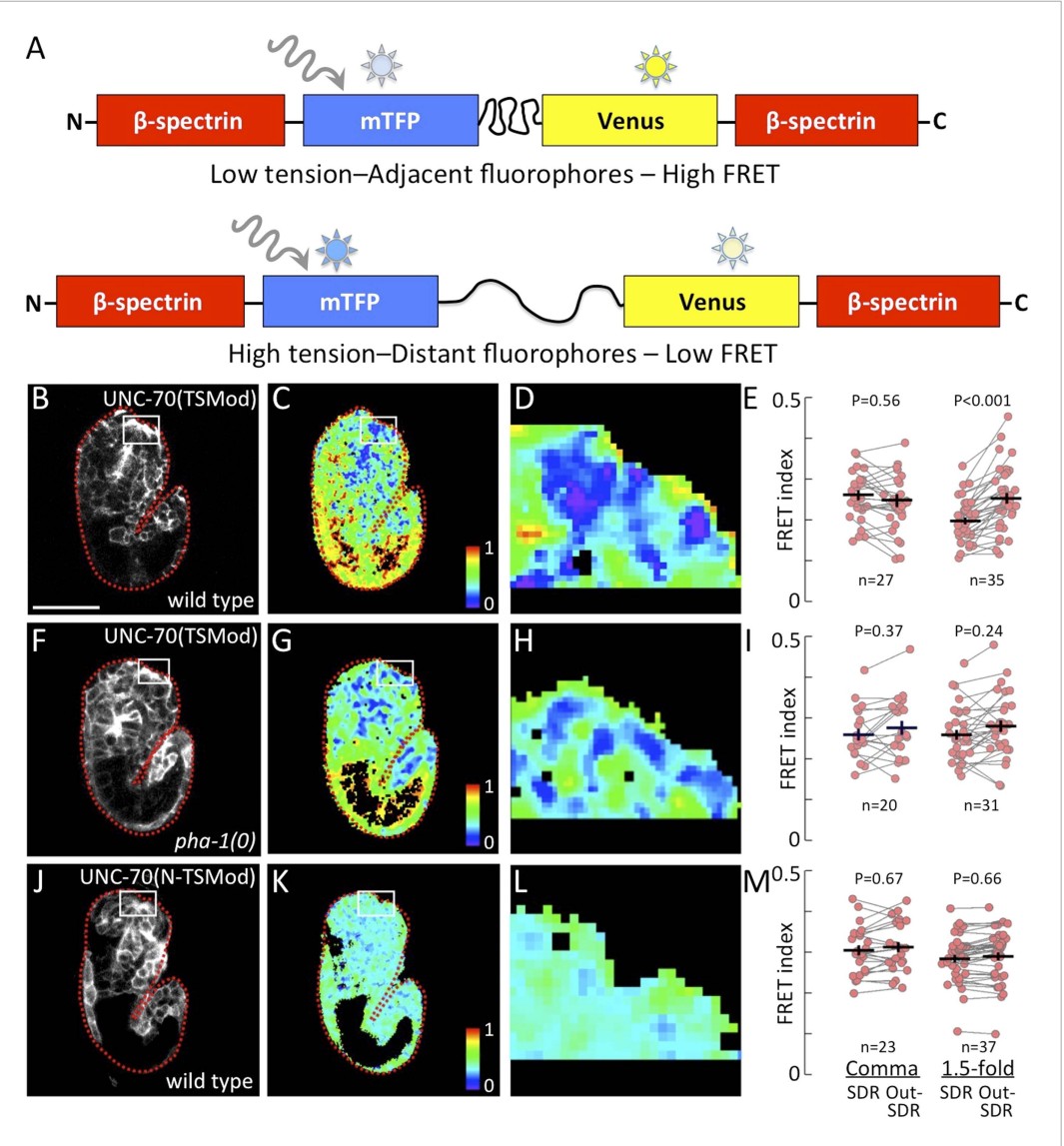

**Figure 3**. Pharyngeal attachment leads to increased forces at sensory depression. A FRET-based TSMod inserted into the *C. elegans* β-spectrin gene (*unc-70*) was used to assess forces in live embryos. (**A**) Schematic for how UNC-70 (TSMod) detects tension. FRET occurs when the donor fluorophore (mTFP) transfers energy to a nearby acceptor fluorophore (Venus) within the same peptide. When UNC-70(TSMod) experiences mechanical tension, a flexible linker separating mTFP and Venus is lengthened, leading to reduced FRET efficiency. (**B–D**) Representative images of wild-type and (F-H) *pha-1*(*tm3671*) strains that express UNC-70(TSMod). (**J–L**) Representative images of wild-type embryos expressing the no force control UNC-70(N-TSMod). Panels **B**, **F**, **J** depict 1.5-fold embryos after direct excitation of the Venus acceptor fluorophore. Panels **C**, **D**, **G**, **H**, **K**, **L** show FRET measurements where purple pixels indicate regions of highest tension (low FRET). Small white-framed boxes in panels **B**, **C**, **F**, **G**, **J**, **K** indicate the sensory depression region (SDR), which is enlarged in panels **D**, **H**, **L**. Red dashed lines in panels **B**, **C**, **E**, **F**, **J**, **K** outline the embryos. Scale bar in B = 30 μm. (**E**, **I**, **M**) FRET indices for the SDR and the region outside the sensory depression (Out-SDR). Individual embryos are represented by red circles, which are connected by lines to indicate values acquired from the same embryo. p-values depicted were calculated using a T-test (also see *Supplementary file 2*). Numbers at the bottom indicate the number of embryos that were analyzed for each condition. Each point is an average of ~3–5 frames from a z-stack encompassing the embryo (see 'Materials and methods' for details).

The following figure supplement is available for figure 3:

**Figure supplement 1**. FRET index of low and high FRET controls.

unlikely to result from differences in expression levels of the sensor or imaging conditions. No significant differences in UNC-70(TSMod) FRET efficiency were observed between the sensory depression region (SDR) and non-SDR region prior to the attachment of the pharynx to the epidermis (early comma stage) in either wild-type or *pha-1(0)* embryos (*Figure 3E,I*). In contrast, wild-type 1.5-fold embryos had significantly higher tension (lower FRET) at the SDR as compared with regions outside the sensory depression (p = 0.0008), consistent with the hypothesis that pharyngeal attachment contributes to the force balance at the sensory depression (*Figure 3C,D,I*; *Supplementary file 2*). In contrast, 1.5-fold *pha-1(0)* mutants in which the pharynx failed to attach did not display appreciably lower levels of mechanical tension at the SDR as compared with regions outside the sensory depression (p = 0.2421; *Figure 3G,H,I*; *Supplementary file 2*). In addition, tension at the SDR was significantly higher in wild-type embryos as compared with the SDR region in *pha-1(0)* mutants at the 1.5-fold (p < 0.0001) but not early comma stages (p = 1.000; *Supplementary file 2*), also consistent with pharyngeal attachment and pulling leading to tension at the anterior epidermis.

To interpret UNC-70(TSMod) FRET signals during early stages of embryogenesis, we replaced the flexible linker by a large separator (TRAF) or a short linker of only five residues (5aa) to generate UNC-70(TRAF) and UNC-70(5aa) constructs in which the two FRET fluorophores are separated by a constant distance and, importantly, are insensitive to force. As expected, both control sensors localized in a pattern that was indistinguishable from endogenous UNC-70 (data not shown; *Moorthy et al., 2000*), rescued the paralyzed phenotype of *unc-70* adults (data not shown; see 'Materials and methods'), and showed FRET values consistent with the distance of the fluorophores (*Figure 3—figure supplement 1*). In addition, no significant differences were observed between SDR and non-SDR regions in 1.5-fold wild-type embryos using these controls (*Figure 3—figure supplement 1*; *Supplementary file 2*) and their FRET efficiency values are similar to those reported previously (*Borghi et al., 2012*; *Krieg et al., 2014*).

To further confirm that the observed differences in the FRET efficiency of UNC-70(TSMod) reliably report differences in molecular tension in our experimental system, we generated animals that carry an UNC-70(N-TSMod) fusion protein, in which the force sensitive FRET construct has been placed at the N-terminus of full-length unc-70 β-spectrin. In this position, the TSMod is not responsive to force and would be predicted to yield FRET signatures consistent with no-force situations. Similar to the other UNC-70 fusion proteins, UNC-70(N-TSMod) was expressed in a pattern indistinguishable to that of the native UNC-70 protein and the transgene restored locomotion to paralyzed *unc-70* adult animals (data not shown). As expected, FRET values were higher in embryos that expressed the force-insensitive UNC-70(N-TSMod) vs UNC-70(TSMod) (*Figure 3*; *Supplementary file 2*), consistent with previous results that a terminal TSMod fusion cannot be pulled apart by cellular forces (*Grashoff et al., 2010*; *Borghi et al., 2012*; *Conway et al., 2013*; *Krieg et al., 2014*). Importantly, we did not see gross variations in FRET across different tissues within the same embryo in N-TSMod expressing animals, consistent with the idea that the variation in UNC-70(TSMod) is due to different forces acting on UNC-70. We also noted that FRET values were independent of the expression level of the fluorophores, indicating that the FRET signal in each pixel was predominantly coming from intramolecular as opposed to intermolecular energy transfer (data not shown). Taken together, the FRET tension sensor provides strong independent support for our model in which the anterior epidermis experiences a high level of mechanical stress that is due in large part to forces exerted by the pharynx (*Figure 1C*).

## MEC-8 regulates the splicing of FBN-1, a fibrillin-like protein

We hypothesized that MEC-8, an RNA-binding protein and known splicing factor (*Lundquist et al., 1996*; *Spike et al., 2002*; *Calixto et al., 2010*), may regulate the mRNA processing of one or more genes that function to stabilize the epidermis in response to mechanical forces. Because the RNA recognition site for MEC-8 is unknown, we used a non-biased approach to identify candidate MEC-8 targets. mRNAs obtained from wild-type and *mec-8* mutant embryos were analyzed using a whole-genome tiling-array approach (*Mockler et al., 2005*; *He et al., 2007*). We identified 1106 individual regions within a total of 449 genes that were differentially expressed (>1.5-fold) between wild-type and *mec-8* embryos (*Supplementary file 3*). This included 159 genes (666 regions) in which at least one exon was upregulated in *mec-8* mutants, 286 genes (421 regions) in which at least one exon was downregulated in *mec-8* mutants and 12 genes (19 regions) in which at least one intron was upregulated in *mec-8* mutants (*Supplementary file 3*). We note that seven genes included in the

totals above contained both upregulated introns and exons. Among the 449 identified genes, 135 (30%) are annotated by WormBase as having multiple (alternatively spliced) isoforms (*Supplementary file 3*). This included 67% (8/12) of the genes with up-regulated introns, 47% (75/159) of genes with up-regulated exons and 22% (52/286) of genes with down-regulated exons. Tiling-array findings were confirmed for several genes within each of the categories described above by PCR analysis (*Figure 4—figure supplement 1*; *Supplementary file 3*).

Many of the identified genes, particularly those with only a single identified mRNA isoform, are unlikely to be direct targets of MEC-8, which regulates alternative splicing (*Spike et al., 2002*; *Calixto et al., 2010*). Such genes are more likely to display transcriptional misregulation as an indirect consequence of *mec-8* loss of function. Also, a significantly higher proportion of the identified genes containing either up-regulated exons or introns were alternatively spliced, as compared with genes containing down-regulated exons (p < 0.0001 and p < 0.005, respectively) or in comparison with all annotated *C. elegans* genes (p < 0.0001 and p < 0.005, respectively; ~25% of *C. elegans* genes are thought to be alternatively spliced; *Ramani et al., 2011*). Given the established role of MEC-8 in alternative splicing, these genes are more likely to include direct targets of MEC-8. This is supported by the observation that *unc-52*, a known target of MEC-8 (*Spike et al., 2002*), was among the exon-up genes identified by the array study and because a second established target of MEC-8, *mec-2*, requires MEC-8 for the removal of one of its introns (*Calixto et al., 2010*). Given that *mec-2* did not, however, meet all of our imposed criteria for designation as a positive outcome from the tiling array, our final gene list is likely to be missing at least some authentic MEC-8 targets.

To identify downstream targets of MEC-8 that are relevant to the synthetic phenotype of *mec-8*; *sym-3* and *mec-8*; *sym-4* mutants, we screened ~200 of the most highly misregulated genes within the dataset for enhancement of the Pin phenotype in single-mutant backgrounds (i.e., *sym-3*, *sym-4* and *mec-8*) using RNAi feeding methods. Although several gene inactivations caused low-to-moderate levels of Pin in one or more of the mutant backgrounds (data not shown), one gene, *fbn-1* (ZK783.1), led to strong enhancement of Pin in both non-RNAi-sensitized and RNAi-hypersensitive mutant backgrounds (see below). In addition, several features of *fbn-1* made it an attractive candidate as a MEC-8 target. In particular, *fbn-1* is notable in that it is one of only 12 genes within the intron-up category, and, based on fold changes, is the third most highly misregulated gene in the tiling array data set (*Supplementary file 3*). Based on the tiling array, the region of *fbn-1* that is misregulated in *mec-8* mutants spans exons 14–19, which includes the region of *fbn-1* that is alternatively spliced (exons 14–16; *Figure 4A,B*). Most notably, expression of an *fbn-1* cDNA (e isoform) driven by native *fbn-1* promoter sequences partially rescued the synthetic lethality of *mec-8*; *sym-4* mutants in two independent lines (*Figure 1B*). This latter finding indicates that *fbn-1* is a critical target for misregulation in *mec-8*; *sym-4* mutants.

To confirm the tiling array results for *fbn-1*, we used PCR to amplify regions of *fbn-1* from cDNA pools derived from wild-type and *mec-8* mutant embryos. Whereas primers amplifying the region spanning exons 14–19 generated multiple bands of the approximate expected sizes in wild type, these bands were either absent or strongly reduced in *mec-8* mutants and were replaced by higher-molecular-weight, or otherwise aberrant, species (*Figure 4A,B*). Consistent with a reduction in splicing efficiency, splicing between exons 16 and 17 was largely abolished in *mec-8* mutants (*Figure 4A,B*). In contrast, splicing between exons 19 and 20 was unaffected in *mec-8* mutants, consistent with both the tiling array findings and the absence of known alternative splicing events between these exons (*Figure 4A,B*). Thus MEC-8 is required for normal splicing events within the region encompassing exons 14 through 19 of *fbn-1*. The observed splicing defects of *fbn-1* mRNA in *mec-8* mutants should result in a reduction in the abundance of wild-type FBN-1 isoforms and reduced FBN-1 activity. In addition, the presence of stop codons within introns 17 and 18 may lead to the production of abnormal truncated forms of FBN-1 (*Figure 4C*). It is also possible that some of these aberrant transcripts are targeted for degradation by RNA surveillance systems that recognize abnormally long non-coding regions within mRNAs (*Mango, 2001*).

## *fbn-1* encodes a protein that shares some domains with vertebrate fibrillins

FBN-1 is composed of many calcium-binding and non-calcium-binding EGF-like repeats, which are found in many matrix proteins and the extracellular domains of transmembrane proteins (*Figure 4C*, *Figure 4—figure supplement 2*; *Davis, 1990*). Comparison of the predicted FBN-1 peptide sequence

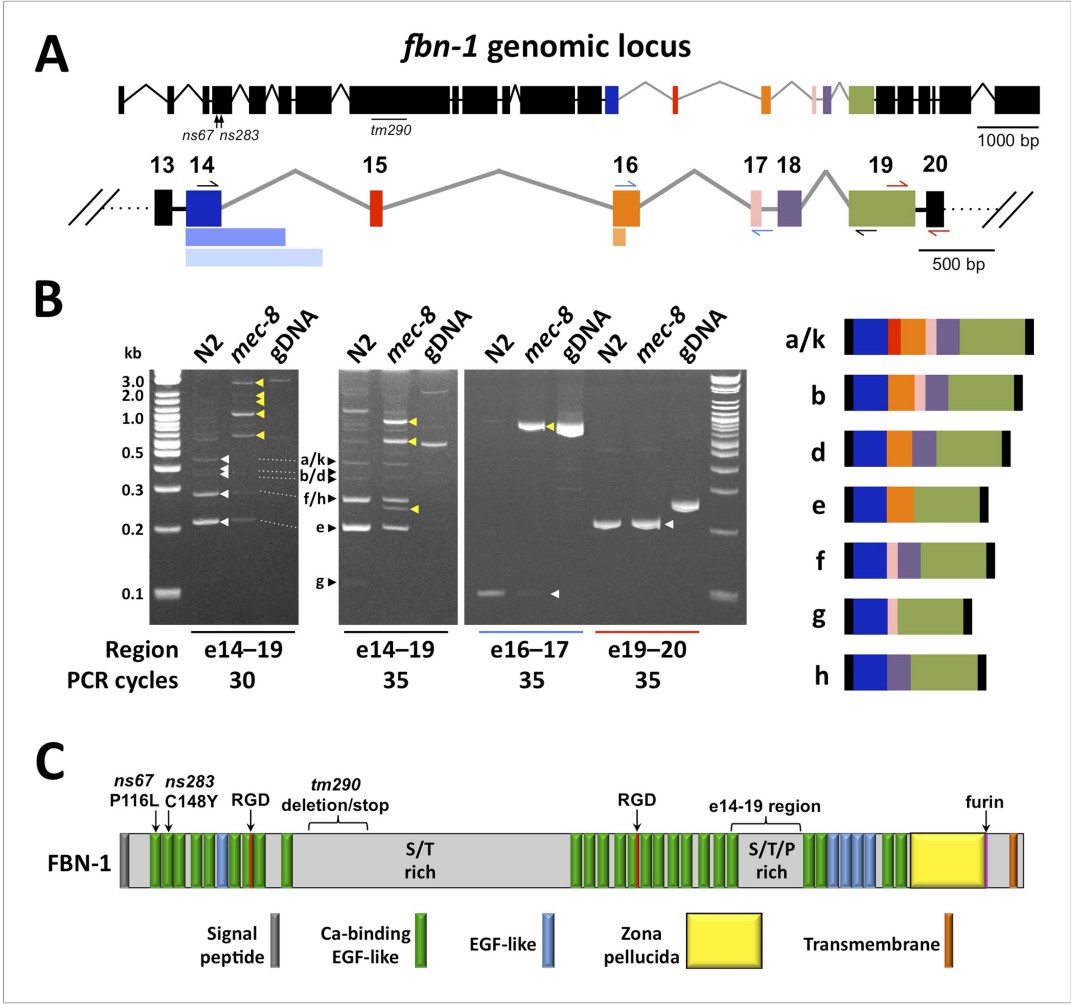

**Figure 4**. Splicing between a subset of *fbn-1* exons is strongly misregulated in *mec-8* mutants. (**A**) A schematic of the *fbn-1* genomic locus is shown with alternatively spliced exons (e14–19) indicated by colored blocks and enlarged below. Single-sided arrows indicate PCR primers used in panel **B**. Lighter-shaded rectangles below exons 14 and 16 indicate alternative 3′ splice sites for these exons. (**B**) PCR of the indicated regions of *fbn-1* using wild-type (N2) and *mec-8* cDNAs derived from embryos and wild-type genomic DNA (gDNA) as templates. White and black arrowheads indicate bands that correspond to known *fbn-1* isoforms (depicted on right) based on size estimations for PCR products (in basepairs): a/k = 476, b = 407, d = 341, e = 200, f = 248, g = 107, h = 182. Yellow arrowheads indicate aberrant *fbn-1* mRNA products that are present or are strongly enriched only in *mec-8* mutants. (**C**) Schematic of FBN-1 (a isoform) showing the locations of protein domains and the amino acid positions affected by *fbn-1* mutant alleles. For an annotated amino acid sequence, see *Figure 4—figure supplement 2*.

The following figure supplements are available for figure 4:

**Figure supplement 1**. Examples of gene regions differentially expressed in *mec-8* mutants and confirmed by RT-PCR.

**Figure supplement 2**. Amino acid sequence of FBN-1.

with mammalian sequences revealed greatest sequence similarity with the family of latent TGFβ binding proteins (LTBPs), which include LTBP-1, -2 and -4 and fibrillins 1–3 (*Rifkin, 2005*; *Todorovic et al., 2005*; *Hynes, 2009*). These proteins carry out structural roles in the ECM in association with elastic fibrils, mediate cell-ECM adhesion, and act as sinks or reservoirs for TGFβ ligands, thereby modulating signal transduction. *C. elegans* FBN-1 differs from the other LTBP proteins by lacking the TGFβ binding domains and by having a zona pellucida (ZP) domain. ZP domains are found in many apical ECM

proteins and are thought to mediate polymerization, resulting in the formation of protein fibrils (*Plaza et al., 2010*). The presence of a furin cleavage site in FBN-1 immediately after the ZP domain suggests that the extracellular domain of FBN-1 can be secreted (*Figure 4C*; also see below). FBN-1 also contains an 834-amino-acid region (560–1393) that is enriched for serine (13%) and threonine (13%) residues as well as a 179-amino-acid region (1920–2098) that is enriched for serine (15%), threonine (24%) and proline (13%) residues. More generally, the sequence of FBN-1, as well as its sequence similarity to vertebrate LTBPs, is consistent with FBN-1 carrying out a structural function in extracellular matrices affiliated with epithelial cells. We note that mis-splicing within the region encompassed by exons 14–19 in *mec-8* mutants should not disrupt any known protein motifs (*Figure 4C*). Nevertheless, this region is well conserved within FBN-1 orthologs in other *Caenorhabditis* family members and is also present in more distantly related parasitic species. In addition, a failure to splice out intron 17 or 18 would lead to a frameshift in the message and a truncated protein that lacks the ZP domain and transmembrane segment.

## FBN-1 functions in a network with MEC-8, SYM-3 and SYM-4 to stabilize epidermal architecture

Three mutant alleles of *fbn-1* were obtained for analysis including two point mutations (*ns67* and *ns283*; M Heiman and S Shaham, unpublished data) and a deletion mutation (*tm290*) generated by the *C. elegans* deletion mutant consortium (*C. elegans Deletion Mutant Consortium, 2012*). Both point mutations lead to non-conservative missense mutations, P116L and C148A, within the first and second EGF-like repeats, respectively (*Figure 4A,C*, *Figure 4—figure supplement 2*). The deletion mutation is missing 604 bp within the eighth exon of *fbn-1*, which encodes a sequence that is serine and threonine rich (*Figure 4A,C*). The *tm290* mutation should produce a protein containing the first 714 amino acids of FBN-1 followed by 224 novel amino acids before encountering a stop codon; the *tm290* transcript may also be targeted for degradation by the non-sense mediated decay pathway (*Mango, 2001*). Whereas the *ns67* and *ns283* alleles were able to be propagated as homozygotes, *tm290* homozygotes were not easily propagated and often arrested during the larval molts, consistent with a previous report (*Frand et al., 2005*).

We first examined *fbn-1* alleles for the presence of the keyhole structure in embryos and the Pin phenotype in L1 larvae. Strikingly, strains containing either missense allele *ns67* or *ns283* exhibited the Pin phenotype in ~45% of their progeny, whereas *tm290* homozygotes produced by homozygous mothers carrying a rescuing *fbn-1*(+) extrachromosomal array had a lower percentage of Pin larvae (~20% within the population of array-minus progeny; *Figure 5B,C*; *Table 1* and *Supplementary file 1*). Consistent with these findings, *fbn-1*(*RNAi*) feeding of RNAi-hypersensitive mutants gave rise to ~30% Pin larvae (*Figure 5A*). In addition, all three alleles of *fbn-1* led to formation of the keyhole in embryos (*Figure 5C*, *Table 1*, *Supplementary file 1*, data not shown). Although the penetrance of Pin in *fbn-1* single mutants was lower than *mec-8*; *sym-3* or *mec-8*; *sym-4* double mutants, the depth of the keyhole observed in some *fbn-1*(*tm290*) homozygous embryos exceeded that observed in *mec-8*; *sym-3* or *mec-8*; *sym-4* embryos (*Figure 5C*, *Table 1*). Thus inhibition of *fbn-1* alone can lead to a compromised embryonic sheath, making the underlying epidermis more susceptible to deformation by mechanical forces including the pulling force exerted by the pharynx. In addition, because *mec-8* homozygous animals are viable and showed a relatively low percentage of Pin larvae (*Figure 1*; *Table 1* and *Supplementary file 1*), we can infer that *fbn-1* function is only partially impaired in *mec-8* mutants, consistent with our tiling array and PCR-based analyses (*Figure 4B*).

We next constructed double mutants between *fbn-1* and *sym-3*, *sym-4* and *mec-8* using the *ns67* and *tm290* alleles. The percentage of Pin animals in double mutants ranged from 97–100%, consistent with the enhancement observed for *fbn-1*(*RNAi*) in RNAi-hypersensitive backgrounds (*Figure 5A–C*, *Supplementary file 1*). In addition, the average depth of the keyhole in these embryos was typically greater than that observed for *mec-8*; *sym-3* or *mec-8*; *sym-4* mutants as well as for *fbn-1* single mutants (*Table 1*). Notably, certain double–mutant combinations displayed phenotypes that had not been previously observed in *mec-8*; *sym-3* or *mec-8*; *sym-4* mutants or in *fbn-1* single mutants. In the case of *fbn-1*(*tm290*); *sym-3* and *fbn-1*(*tm290*); *sym-4* mutants, large lumps or protuberances on the head region were observed in L1 larvae (*Figure 5C*), which are reminiscent of certain phenotypes observed in integrin pathway mutants (*Baum and Garriga, 1997*; *Tucker and Han, 2008*).

Interestingly, *mec-8*; *fbn-1*(*tm290*) mutants arrested uniformly as embryos and failed to complete embryonic elongation (*Figure 5C*). These embryos displayed a deep keyhole by the 1.5-fold stage

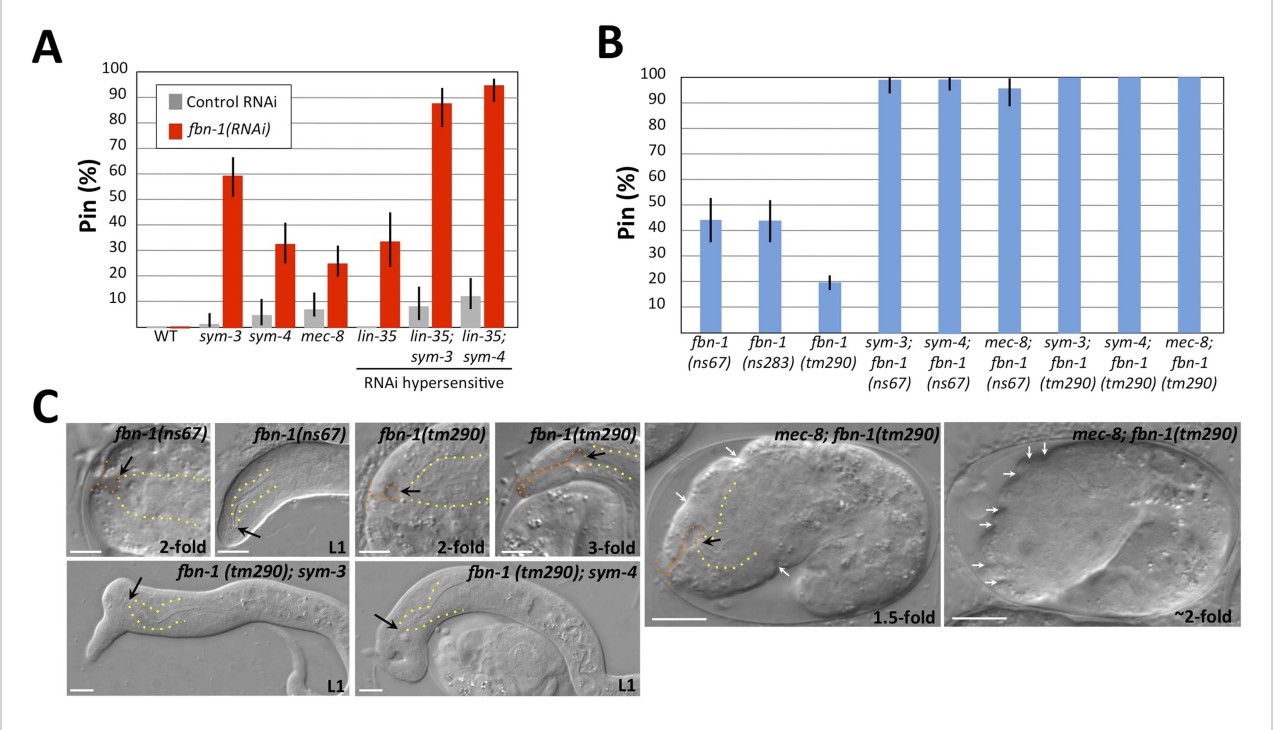

**Figure 5**. Morphogenesis defects of *fbn-1* mutants are strongly enhanced by mutations in *sym-3*, *sym-4* and *mec-8*. (**A**) RNAi feeding of *fbn-1* was carried out in the indicated backgrounds including strains hypersensitized to RNAi. Control RNAi strains contained the vector plasmid pPD129.36. (**B**) The Pin phenotype was scored in *fbn-1* mutant alleles and in selected double mutants with *fbn-1* and *mec-8*, *sym-3* or *sym-4*. Error bars in **A** and **B** represent 95% CIs. For additional information, see *Table 1* and *Supplementary file 1*. (**C**) Representative images for select single and compound mutants. Note the presence of strong head malformations in *fbn-1*(*tm290*); *sym-3* and *fbn-1*(*tm290*); *sym-4* larvae. Also note that the strong epidermal malformations observed in *fbn-1*(*tm290*); *mec-8* mutants are suppressed by *let-502*(*RNAi*). White arrows indicate ingressions or furrows throughout the epidermis; red arrows, detached anterior cells in *fbn-1*(*tm290*); *mec-8* mutants. Yellow dashed lines indicate lateral pharyngeal borders; orange dashed lines, the sensory depression or keyhole; black arrows, posterior extent of ingression. White scale bars = 10 μm.

(*Table 1* and *Supplementary file 1*), and by the ∼threefold stage 92% (n = 73) exhibited prominent epidermal ingressions and furrows, which were often regularly spaced (*Figure 5C*). Notably, this phenotype was previously observed after digestion of the embryonic sheath with trypsin (*Priess and Hirsh, 1986*), suggesting that FBN-1 carries out mechanostructural functions throughout the embryonic sheath including a role in stabilizing the epidermis during circumferential constriction. Consistent with this interpretation, inhibition of epidermal actomyosin contraction using *let-502*(*RNAi*) reduced the frequency of *mec-8*; *fbn-1*(*tm290*) embryos that contained deep furrows to 17% (n = 52; *Figure 5C*). We note that in addition to surface furrows and blobs, some *mec-8*; *fbn-1*(*tm290*) embryos also showed cell detachment phenotypes (*Figure 5C*), suggesting that MEC-8 and FBN-1 promote epidermal integrity. Because *tm290* is likely to constitute a null mutation in *fbn-1*, we interpret the severe phenotype of *mec-8*; *fbn-1*(*tm290*) mutants to indicate that MEC-8 regulates additional proteins that act redundantly with FBN-1 together to promote normal epidermal structure and morphogenesis.

### Activity of *fbn-1* is required in the epidermis

On the basis of the above findings, we hypothesized that FBN-1 is a component of the embryonic sheath, a specialized ECM secreted from the apical surface of epidermal cells that promotes structural stability and resistance to biomechanical forces (*Priess and Hirsh, 1986*). A requirement for *fbn-1* in the epidermis was first tested by treating wild-type and NR222 strains with *fbn-1*(*RNAi*) using standard feeding methods. Whereas wild-type strains can undergo 'systemic' RNAi (throughout the majority of tissues), NR222 is engineered to undergo RNAi in the epidermis only (*Qadota et al., 2007*) (*Figure 5A*, *Figure 6A*, *Supplementary file 1*). Although RNAi of *fbn-1* failed to produce any visible phenotype in these strains, enhancement of the Pin phenotype was observed in an NR222 derivative

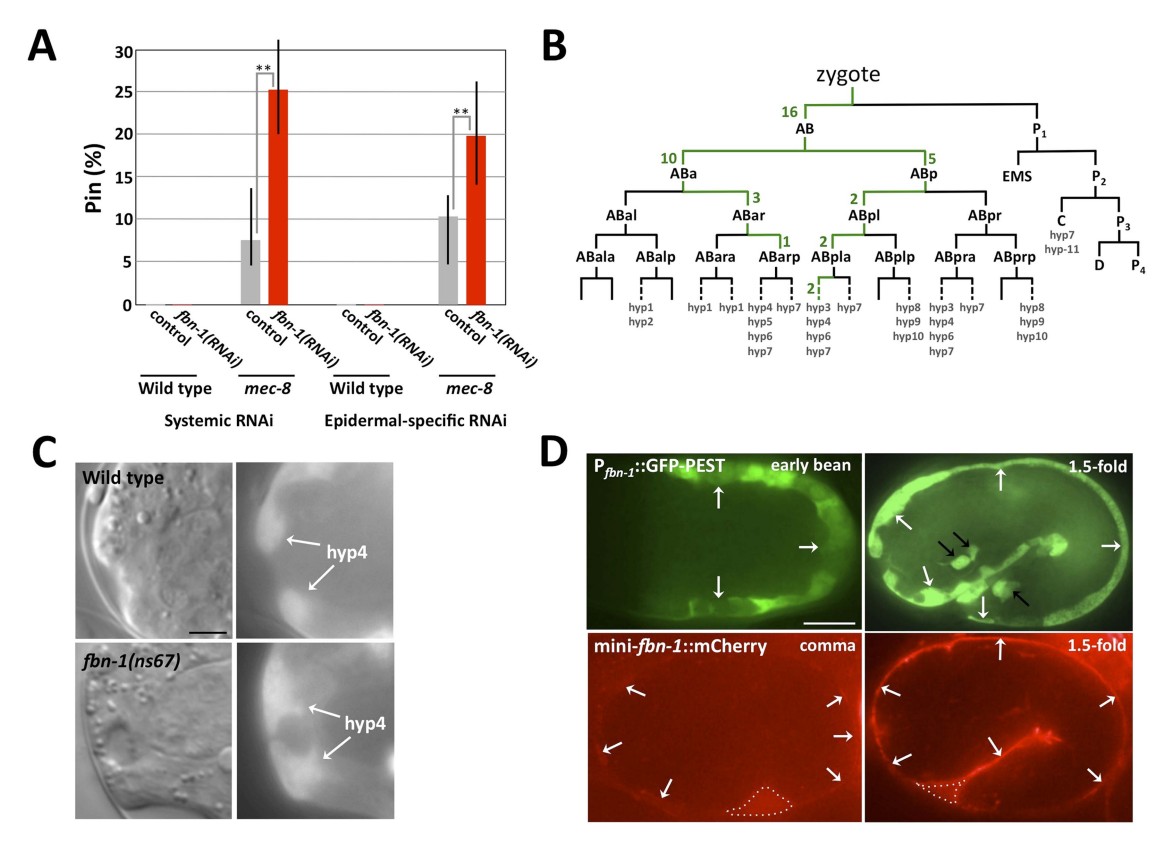

Figure 6. The *fbn-1* gene is required in the epidermis and specifies a component of the embryonic sheath. (**A**) Systemic and epidermal-specific RNAi of *fbn-1* was carried out in wild-type (N2) and strain NR222, respectively, and in both backgrounds containing the *mec-8(u74)* allele. Note that both systemic and epidermal-specific *fbn-1(RNAi)* led to an increased percentage of Pin animals in the *mec-8* background. Error bars indicate 95% CIs; **p < 0.01. (**B**) Schematic of the *C. elegans* embryonic lineage and findings from the *fbn-1* mosaic analysis. Strains used for the analysis were WY1059, *fbn-1(tm290)*; *sym-4(mn619)*; *fdEx249[fbn-1(+)*; *sur-5::GFP]*, and WY1068, *mec-8(u74)*; *fbn-1(tm290)*; *fdEx249*. Green numbers indicate the number of L4 or adult mosaic animals that were not Pin but contained the *fbn-1(+)* rescuing array within that lineage only. (**C**) Wild-type and *fbn-1(ns67)* embryos expressing P*fbn-1*::GFP-PEST (a convenient marker for embryonic epidermal cells). hyp4 cells within the focal plane are indicated and show aberrant morphologies in mutant embryos that contain a keyhole. (**D**) Expression of P*fbn-1*::GFP-PEST and mini-*fbn-1*::mCherry (Δ*fbn-1*–49-2418) reporters. In the P*fbn-1*::GFP-PEST panels, epidermal cells are indicated with white arrows. Black arrows indicate several cells positive for P*ttx-3*::GFP, which was used as an injection marker and is not expressed in epidermal cells. In the mini-*fbn-1*::mCherry panels, the apical surface of embryonic epidermal cells (sheath) is indicated by white arrows. mini-*fbn-1*::mCherry is also detected in the extra-embryonic space (white dashed triangles). White scale bar = 10 μm, black bar = 5 μm.

that carried a *mec-8* mutation (p < 0.01), implicating the epidermis as critical for *fbn-1* activity (*Figure 6A* and *Supplementary file 1*).

Genetic mosaics (*Yochem and Herman, 2005*) were also examined for the focus of *fbn-1* activity in the prevention of the Pin phenotype. This non-biased approach used an *fbn-1(tm290)*; *sym-4(mn619)* strain carrying a rescuing *fbn-1(+)* extra-chromosomal array, *fdEx249*, that also expresses a fluorescent reporter, *sur-5::GFP*, to assess mitotic inheritance of the array (*Yochem et al., 1998*). This strain segregated array-minus Pin progeny, array-plus viable progeny and array-plus viable progeny that were mosaic for inheritance of the array; array-minus non-Pin *fbn-1(tm290)*; *sym-4(mn619)* animals were not observed. Based on numerous mosaics, *fbn-1* activity is focused in hyp6, the anterior portion of the hyp7 syncytium, or both hyp6 and hyp7. An exact determination of inheritance was not possible because both the hyp6 and hyp7 syncytia initiate formation through cell fusion near the time the keyhole (Pin) becomes apparent and the hyp6 syncytium fuses with the hyp7 syncytium late in the L2 stage (*Yochem et al., 1998*). Nevertheless, two mosaics proved particularly informative in that hyp7 was the only positive tissue. Moreover, SUR-5::GFP was expressed in an anterior-to-posterior gradient in both mosaics, suggesting establishment of the positive clone within the hyp7 syncytium by

one or more hyp6 cells, which are anterior, or possibly one or more anterior hyp7 cells. Additional mosaics were consistent with a requirement for *fbn-1*(+) in anterior epidermal cells. For example, in 16 mosaics, only AB, one of the daughters of the zygote, had established a positive clone (*Figure 6B*). In contrast, there were no reciprocal mosaics in which P1, but not AB, had established a positive clone. Although 12 of the hyp7 cells of the embryo descend from $P_1$, these cells are not located as far anterior in the embryo as are certain hyp7 (or hyp6) cells that descend from AB (*Sulston et al., 1983*). In addition to these 16 AB(+)$P_1$(−) mosaics, there were 25 mosaics in which positive clones had been established within the AB sublineage only. In every case, these clones contributed descendants to hyp6 or to the anterior part of hyp7 (*Figure 6B*).

Although the genetic mosaics are consistent with the epidermal-specific RNAi described above, the mosaics cannot eliminate other anterior epidermal cells as important for expression of *fbn-1*. For example, although hyp4 cells, which are closer to the sensory depression than hyp6 or hyp7 cells, were not specifically implicated in the analysis, they could still contribute significant FBN-1 for proper function of the sheath in wild-type embryos. For example, a contribution by hyp4 could be obviated in mosaics by over-expression of *fbn-1* in hyp6 or anterior hyp7 cells, particularly if it is diffusible following secretion from the apical surface of these cells. In fact, a requirement for *fbn-1* in the sheath surrounding hyp4 is consistent with the observed deformation of hyp4 cells in *fbn-1* mutants (*Figure 6C*). Neither sheath nor socket cells associated with the sensory depression were implicated in the mosaic analysis, underscoring the requirement for *fbn-1* expression in the epidermis for the prevention of Pin. Also of note, the molting defect associated with *fbn-1*(*tm290*) was rescued in all of the non-Pin mosaics. Thus, the epidermis appears to be the sole focus for both major aspects of the *fbn-1* phenotype.

## FBN-1 is expressed in embryonic epidermal cells and secreted to the apical surface

To more directly assess *fbn-1* expression in live embryos, we used strains that contained one of two *fbn-1* fluorescent reporters. P*fbn-1*::GFP-PEST is expressed under the control of the native *fbn-1* promoter and contains PEST sequences, which reduce the half-life of GFP (*Frand et al., 2005*). P*fbn-1*::GFP-PEST expression was first detected in epidermal cells at the onset of embryonic morphogenesis, and expression continued throughout embryogenesis (*Figure 6D*). The mini-*fbn-1*::mCherry reporter includes both an N-terminal region (aa 1–48) that contains a predicted signal peptide (aa 1–26) and a portion of the C terminus (aa 2418–2781) that includes the ZP domain (aa 2438–2674), the furin cleavage site (aa 2676–2679) and the predicted transmembrane segment (aa 2745–2767). mini-*fbn-1*::mCherry localized to the apical surface of epidermal cells coincident with the location of the embryonic sheath (*Figure 6D*). Expression was first detected during early stages of morphogenesis and increased in intensity through the 1.5-fold stage, consistent with the timing of embryonic sheath formation (*Figure 6D*; *Priess and Hirsh, 1986*). mini-*fbn-1*::mCherry was also detected during late stages of embryogenesis and in larvae (to be described elsewhere). Notably, mini-*fbn-1*::mCherry was detected in the extra-embryonic space of early morphogenetic embryos, consistent with apical secretion of the fusion proteins (*Figure 6D*). The expression of FBN-1 in epidermal cells and its secretion to the apical surface is consistent with the model that FBN-1 functions as a structural component of the embryonic sheath where it prevents mechanical deformation of the epidermis.

## The embryonic sheath prevents epidermal deformation by multiple forces

*Priess and Hirsch (1986)* used laser permeabilization of the eggshell followed by trypsin treatment to induce digestion of the embryonic sheath. Although they reported striking indentations or furrows at the surface of ~twofold-stage trypsin-treated embryos, similar to what we observed for *mec-8*; *fbn-1* (*tm290*) mutants (*Figure 5C*), defects of the sensory depression were not described. We therefore carried out a similar experiment in which we used chitinase to partially or completely digest the eggshell followed by trypsin treatment. Most notably, we detected keyholes in ~1.5-fold to 3.0-fold-stage embryos, as well as mild ingressions at the surface of some embryos (*Figure 7A*). We note that the surface ingressions we observed were less dramatic than those reported in the previous study, which may be due in part to the different methods used to permeabilize the eggshell. Epidermal ingressions induced by trypsin treatment were also less severe than those

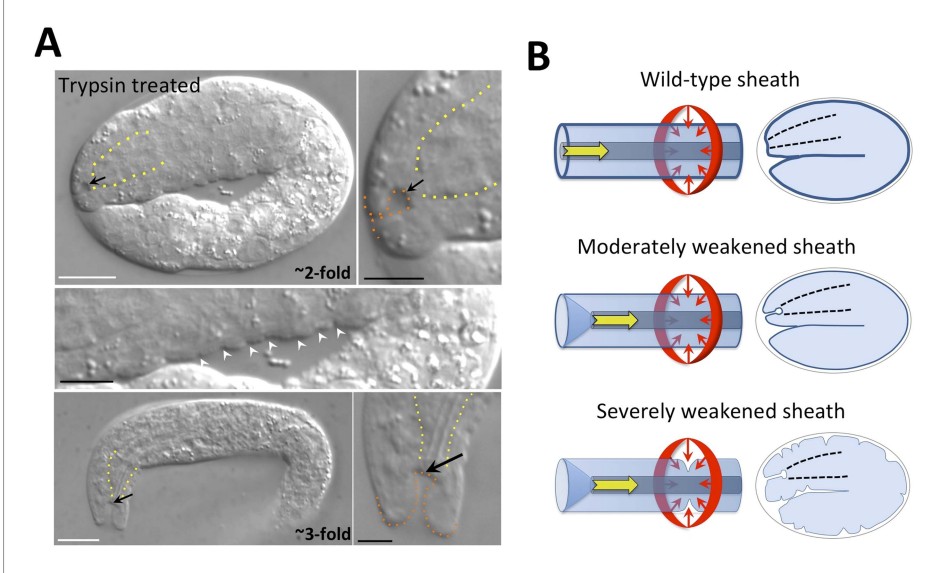

**Figure 7**. The embryonic sheath is critical for resistance to biomechanical forces. (**A**) Wild-type embryos were treated with chitinase to remove part or all of the eggshell and then with trypsin to digest the sheath. Note the presence of a keyhole in both twofold and threefold trypsinized embryos (black arrows) and multiple ingressions or furrows in the epidermis of a twofold embryo (white arrowheads). Yellow dashed lines indicate lateral pharyngeal borders; orange dashed lines, the sensory depression or keyhole. White scale bars = 10 µm, black bars = 5 µm. (**B**) Model for the circumferential squeezing force (red arrows) and pharyngeal pulling force (yellow arrow) that act on the embryonic sheath. When the sheath is moderately weakened, such as when *fbn-1* function is partially impaired, a keyhole phenotype is observed, suggesting that the anterior epidermis is particularly sensitive to a reduction in sheath integrity as a result of the pharyngeal pulling force. In cases where the sheath is more severely compromised, the depth of the keyhole may further increase, and the embryonic epidermis develops ingressions or furrows where circumferential constricting forces are acting.

observed for *mec-8*; *fbn-1*(*tm290*) mutants (*Figure 5C*), suggesting that the sheath was more severely compromised in these double mutants. Of interest, keyholes were seen in some embryos that lacked other obvious morphological defects (data not shown), suggesting that the region of the pharyngeal attachment is particularly sensitive to deformation after partial degradation of the sheath. These findings were also consistent with the lack of gross morphological defects seen in *fbn-1* single mutant embryos as well as *mec-8*; *sym-3* or *mec-8*; *sym-4* double mutants, which nevertheless had a prominent keyhole and Pin phenotype. We note that the pharyngeal cuticle contains the polysaccharide chitin and thus treatment with chitinase could be expected to preferentially degrade the pharyngeal cuticle as well as the eggshell (*Zhang et al., 2005*). Nevertheless, the penetrance of Pin was ~10-fold higher in embryos treated with both trypsin and chitinase relative to chitinase alone (data not shown), consistent with a role for sheath proteins in preventing mechanical deformation of cells surrounding the sensory depression.

## Discussion

Force is essential for shaping the embryo and its internal organs (*Keller et al., 2003*, *2008*; *Davidson, 2011*; *Davidson, 2012*; *Heisenberg and Bellaiche, 2013*), and spatiotemporal application of tightly controlled forces ensures normal morphogenesis. Proper development also requires that cells and tissues that experience forces respond in a consistent and context-appropriate manner. Either too much or too little resistance on the part of targeted tissues can lead to morphogenetic abnormalities and birth defects (*Epstein et al., 2004*; *Moore et al., 2013*).

Our studies have implicated FBN-1, along with MEC-8, SYM-3 and SYM-4, in promoting correct epidermal morphology and resistance of the *C. elegans* epidermis to two biomechanical forces. One force is generated by the circumferential constriction of epidermal actomyosin rings and was identified nearly 20 years ago as the major driver of embryonic elongation (*Priess and Hirsh, 1986*).

We have described here a second force in which the elongating pharynx exerts an inward pull on the anterior epidermis throughout much of embryonic development. Although the potential for a pulling force was suggested by a previous study describing early steps of pharyngeal morphogenesis (*Portereiko and Mango, 2001*), this force was not characterized in any detail. Our studies suggest that this force may result from an intrinsic mechanical resistance of the embryonic pharynx to stretching. Moreover, the epidermal constricting force and pharyngeal stretching force are mechanistically linked because the extension of the pharynx requires the elongation of the epidermis. We also note that whereas inhibition of *fbn-1* alone led to decreased resistance to the pharyngeal pulling force, deformation of the lateral epidermis by the circumferential constricting force required the simultaneous loss of *fbn-1* and *mec-8*, indicating that additional targets of MEC-8 likely contribute to epidermal stability.

Our studies indicate that FBN-1, a protein that is related to fibrillin, is critical for biomechanical force resistance by the epidermis during development. FBN-1 was broadly expressed in the embryonic epidermis and was secreted to the apical surface as a putative component of the embryonic sheath. In Pin embryos that lacked wild-type *fbn-1* activity, progenitor cells of the hyp4 epidermal syncytium became hyperextended. Although hyp4 cells were not directly implicated as the focus for *fbn-1* expression by the mosaic analysis, secreted ECM proteins can rescue defects at a distance or when expressed from cell types that are not normally the source of the protein product (*Heiman and Shaham, 2009*). This is particularly true if proteins are overexpressed, as is often the case for mosaic analysis. The lack of identified mosaic animals in which hyp4 was the only positive epidermal clone may be due to their low frequency of occurrence or because expression of *fbn-1* in hyp4 is not essential for rescue of Pin if other neighboring cells secrete high levels of FBN-1. Alternatively, expression of FBN-1 in hyp6/7 progenitors could possibly alter the biophysical properties of the sheath and the tension on hyp4 cells. More generally, our findings implicate *fbn-1* expression in the anterior epidermis as critical for suppression of the Pin phenotype. In addition, analysis of *mec-8*; *fbn-1* double mutants indicated a role for FBN-1 throughout the embryonic sheath in resisting or properly distributing forces that arise during circumferential constriction of the epidermis.

We failed to detect morphological defects at the attachment site of the intestine to the rectum in any of the mutant backgrounds examined. Although it is possible that stretching of the intestine during embryonic elongation may also lead to forces that act on the posterior epidermis of the worm, such forces may be lower in magnitude than those experienced at the anterior, possibly because of structural differences between the pharynx and intestine.

A number of human diseases that affect ECM components can lead to altered mechanical properties of the skin and other connective tissues, which may parallel defects observed in *C. elegans fbn-1* mutants (*Judd, 1984*; *Milewicz et al., 2000*). This includes mutations in human fibrillin 1, which is mutated in Marfan syndrome (*Dietz et al., 2005*; *Ramirez and Dietz, 2009*; *Ramirez and Sakai, 2010*). Although our findings suggest that FBN-1 and human fibrillins may carry out some related functions in the ECM, structural differences between FBN-1 and vertebrate fibrillins suggest significant functional divergence (*Piha-Gossack et al., 2012*). For example, FBN-1 lacks conserved TGFβ binding sites found in human fibrillins and contains a ZP domain not found in LTBP family proteins. Thus, whereas mammalian fibrillins and other members of the LTBP family of proteins have secondary roles in modulating signal transduction, their closest counterparts in nematodes may be limited to structural functions only. In addition, mammalian fibrillins interact with elastins (*Baldwin et al., 2013*), which are not present in *C. elegans*. A phylogenetic analysis suggested that fibrillins may have been lost or severely disrupted in the nematode lineage as well as in *Drosophila*, although apparent fibrillin orthologs are present in several insect species including ants and honeybees (*Piha-Gossack et al., 2012*).

Our findings in *C. elegans* suggest that FBN-1 is required in the embryonic sheath to ensure the appropriate level of resistance to mechanical deformation by inward-pulling forces (*Figure 7B*), a function originally proposed for the sheath by *Priess and Hirsch (1986)*. This could be because FBN-1 directly affects the resilience of the embryonic sheath, thereby influencing the response of attached epidermal cells to mechanical forces. Alternatively, FBN-1 may be required for the stable attachment of epidermal cells to the sheath or the efficient transmission and distribution of forces throughout the sheath. For example, FBN-1 in the sheath could interact with other transmembrane proteins expressed on the apical surface of the epidermis, consistent with the presence of putative integrin (Arg/Gly/Asp; RGD) binding sites in FBN-1 (*Figure 4—figure supplement 2*). In this latter

scenario, inward-pulling forces may physically detach epidermal cells from the overlying sheath in mutants with compromised *fbn-1* function, leading to excessive or atypical deformation of the unattached epidermis. In addition, it is also possible that FBN-1 could promote attachment of the epidermis to the sheath through its transmembrane domain, although cleavage of FBN-1 by furin proteases, a post-translational processing event conserved in human fibrillins (*Lönnqvist et al., 1998*; *Ashworth et al., 1999*), make this mechanism less likely.

More generally, our results suggest that mechanical properties of the aECM strongly affect epidermal cell architecture and embryonic morphogenesis. Although originally thought to function exclusively as a protective barrier to the environment, the aECM has recently been recognized to play key roles in epithelial morphogenesis, tube formation and cell junction stability (*Hynes, 2009*; *Brown, 2011*). In *C. elegans*, several aECM proteins containing extracellular leucine-rich only (eLLRon) repeats are required to maintain the integrity of epithelial junctions within the lumen of the excretory system (*Mancuso et al., 2012*) and to promote dendrite branching (*Liu and Shen, 2012*). Correspondingly, the *Drosophila* eLLRon protein dALS/convoluted is required to organize the tracheal lumen matrix, and mutations in *dALS* lead to tracheal tube morphogenetic defects (*Beitel and Krasnow, 2000*; *Swanson et al., 2009*). A second class of aECM protein, those containing a ZP domain, have been implicated in tubulogenesis and epithelial morphogenesis in *Drosophila* (*Denholm and Skaer, 2003*; *Jazwinska et al., 2003*; *Roch et al., 2003*; *Plaza et al., 2010*; *Dong et al., 2014*). The *Drosophila* ZP-domain protein DPY has been proposed to organize cuticle architecture and to anchor the cuticle to the epidermis. Based on its size, structural motifs and mutant phenotypes, DPY has been proposed to distribute mechanical tension within the cuticle, thereby stabilizing the attachment of the epidermis to the cuticle (*Wilkin et al., 2000*). Notably, *dpy* is the closest *Drosophila* relative to *C. elegans fbn-1*, and their related mutant phenotypes suggest strong functional conservation. In *C. elegans*, DYF-7, a ZP-domain protein, and DEX-1, which contains a zonadhesin domain, are required to anchor dendrite endings at the nose while the neuronal cell bodies migrate away, stretching the dendrites behind them as they migrate (*Heiman and Shaham, 2009*). These neurons may experience mechanical tension during the process of retrograde dendritic extension, and mutations in *dex-1* or *dyf-7* lead to morphogenetic defects in these neurons and associated glia. In addition to eLLRon and ZP-domain proteins, several other classes of aECM proteins have been implicated in epithelial morphogenesis in *C. elegans*, *Drosophila* and other species (*Lane et al., 1993*; *von Kalm et al., 1995*; *Moussian et al., 2007*; *Willenborg and Prekeris, 2011*; *Labouesse, 2012*; *Syed et al., 2012*; *McLachlan and Heiman, 2013*; *Luschnig and Uv, 2014*). Because our tension sensor records molecular tension only within UNC-70/β-spectrin, it remains unclear how the observed intracellular tension is related to tension in the aECM. Further studies of the mechanical resistance of the embryonic sheath and the forces transmitted through cell-adhesion and matrix-anchoring molecules are needed to elucidate this important aspect of morphogenesis.

Although our studies demonstrate that FBN-1 is an important downstream target of MEC-8 in promoting force resistance by the epidermis, it is clearly not the only target of MEC-8 that carries out structural or biomechanical functions. Cytoskeletal and ECM proteins implicated by the *mec-8* tiling array studies include AJM-1, a component of epithelial adherens junctions (*Köppen et al., 2001*); LET-805, a fibronectin repeat protein (*Hresko et al., 1999*); UNC-52/perlecan, a component of basement membranes (*Rogalski et al., 1993*); VAB-10/plakin a cytoskeletal crosslinker (*Bosher et al., 2003*) and UNC-70/β-spectrin (*Hammarlund et al., 2000*). Based on our genetic data, misregulation of *fbn-1* may largely account for the role of *mec-8* in the context of its synthetic phenotype with *sym-3* or *sym-4*, although one or more additional MEC-8 targets may contribute to the anterior epidermal defects of *mec-8*; *sym-3* or *mec-8*; *sym-4* double mutants. Furthermore, the synthetic embryonic lethality observed in *mec-8*; *fbn-1(tm290)* double mutants (*Figure 5C*) indicates that MEC-8 regulates the splicing of one or more genes that function redundantly with FBN-1.

Similar to *C. elegans mec-8*, mutations in the *Drosophila mec-8* ortholog, *coach potato* (*cpo*), lead to neuronal and behavioral defects (*Perkins et al., 1986*; *Bellen et al., 1992a*, *1992b*; *Glasscock and Tanouye, 2005*), although the splicing targets of Cpo are unknown. In addition, *cpo* is implicated in diapause regulation and climatic adaptation through an unknown mechanism (*Schmidt et al., 2008*). RBPMS and RBPMS2, the human orthologs of MEC-8, are broadly expressed but very little is known about their targets or biological functions (*Shimamoto et al., 1996*). Interestingly, human *FBN1* and *FBN3* are alternatively spliced and distinct *FBN1* isoforms are expressed in a tissue and developmental-specific manner (*Corson et al., 1993*, *2004*; *Biery et al., 1999*; *Burchett et al., 2011*). Furthermore,

alternative splicing of *FBN1* has been suggested to modulate the severity of Marfan syndrome (*Burchett et al., 2011*). Although it is tempting to speculate that RBPMS could be a candidate regulator of human fibrillins, it must be noted that the region of *fbn-1* that is regulated by MEC-8 (e14–e19) is not conserved outside of nematodes nor is the RNA recognition sequence for MEC-8/Cpo/RBPMS currently known.

How SYM-3 and SYM-4 promote epidermal stability or ECM maintenance is at present unresolved. SYM-4 is a predicted β-propeller protein with seven WD-repeats, suggesting a role in coordinating protein interactions. Two independent groups identified mammalian SYM-4, WDR44, as a binding partner and candidate effector of the Rab11 GTPase (*Mammoto et al., 1999*; *Zeng et al., 1999*). WDR44 associates specifically with the activated GTP-bound form of Rab11 and partially co-localizes with Rab11 (*Mammoto et al., 1999*; *Zeng et al., 1999*). Rab11 has been studied in multiple contexts and is primarily associated with the regulation of trafficking to and from the endocytic recycling compartment (*Urbe et al., 1993*; *Ren et al., 1998*; *Grant and Donaldson, 2009*; *Horgan et al., 2010*; *Kelly et al., 2012*) but also functions in exocytosis and in conjunction with the exocyst complex (*Chen et al., 1998*; *Satoh et al., 2005*; *Ward et al., 2005*; *Sato et al., 2008*; *Takahashi et al., 2012*; *Welz et al., 2014*) and in Golgi-endosome transport (*Ullrich et al., 1996*; *Wilcke et al., 2000*). In *C. elegans*, RAB-11 regulates endosomal recycling during mitosis (*Blethrow et al., 2004*; *Ai et al., 2009*), cytokinesis (*Bembenek et al., 2010*) and meiosis (*Cheng et al., 2008*) and, most notably, promotes secretion and ECM formation in embryos (*Sakagami et al., 2008*; *Wehman et al., 2011*). Based on a high-throughput screen, the *Drosophila* SYM-4 ortholog, CG34133, physically interacts with Amph/Amphiphysin (*Guruharsha et al., 2011*), a BAR-domain protein that promotes endocytosis through membrane bending and vesicle fission (*Peter et al., 2004*; *Campelo and Malhotra, 2012*; *Cowling et al., 2012*), suggesting that SYM-4 may interact directly with components of the vesicular trafficking machinery.

SYM-3 contains an N-terminal C2 domain (NT-C2/EEIG1/EHBP1), which suggests an association with the cytoplasmic surface of cell membranes (*Zhang and Aravind, 2010*). Intriguingly, the physical interaction between WDR44 and Rab11 was previously proposed to require an unidentified membrane-associated factor (*Zeng et al., 1999*). The only other *C. elegans* NT-C2 protein, EHBP-1, is a co-partner of RAB-10 in endocytic recycling (*Shi et al., 2010*) and an NT-C2 domain is present in the mammalian Rab11 interactor, Rab11-FIP2 (*Hales et al., 2002*; *Welz et al., 2014*). The *Drosophila* SYM-3 ortholog, CG8671, is required for efficient dsRNA uptake, a process that requires receptor-mediated endocytosis (*Saleh et al., 2006*). Correspondingly, *sym-3* inhibition may lead to a modest reduction in the sensitivity of *C. elegans* to RNAi feeding (*Saleh et al., 2006*). Thus, although their specific molecular functions are largely uncharacterized, available evidence points to a role for both SYM-3 and SYM-4 in vesicular trafficking and endocytosis and/or endocytic recycling.

We propose that SYM-3 and SYM-4 may co-regulate the cell-surface trafficking of one or more proteins that regulate epidermal stability. Loss of *sym-3* or *sym-4* activity could potentially result in the mislocalization of one or more integral membrane proteins or ECM components required for normal resistance to mechanical stress. Correspondingly, the combined loss of both *mec-8* and either *sym-3* or *sym-4* activity most likely lead to a synergistic effect on the epidermis and the observed synthetic phenotype. SYM-3 and SYM-4 may regulate the secretion of aECM proteins, such as FBN-1, or may control the trafficking of integral membrane proteins required for the adhesion of epidermal cells to the aECM or possibly other cell types. We note that the lack of any molting defect in *mec-8*; *sym-3* and *mec-8*; *sym-4* mutants, a phenotype observed following strong loss of function of *fbn-1*, is perhaps most consistent with SYM-3 and SYM-4 acting on a target distinct from FBN-1. In any case, the roles of MEC-8, SYM-3 and SYM-4 in morphogenesis are revealed only under genetic conditions in which overlapping or redundant functions are inhibited. Further studies to fully understand the basis for morphogenesis and the role of the aECM in development are also likely to require approaches that address and overcome limitations imposed by genetic redundancy.

## Materials and methods

### Strains and maintenance

All strains were cultured on nematode growth medium (NGM) supplemented with *Escherichia coli* OP50 as a food source according to standard protocols (*Stiernagle, 2006*) and were maintained at 20°C except for strains containing the *sqt-3(e2117)* allele. Strains used in this study included N2,

SP2231 [*sym-3(mn618) X*], SP2232 [*sym-4(mn619) X*], WY893 [*mec-8(u74) I*; *sym-3(mn618) X*; *mnEx169 (sym-3(+);sur-5::GFP*)], WY969 [*mec-8(u74) I*; *sym-4(mn619) X*; *fdEx226 (sym-4(+)*; *sur-5:GFP*; *P_{hsp-16}::peel-1*)], SP1750 [*mec-8(u74) I*; *mnEx2 (mec-8(+)*; *pRF4rol-6(su1006d)*)], WY870 [*mec-8 (u74) I*; *pha-1(e2123ts) III*; *sym-3(mn618) X*; *mnEx169*], WY873 [*mec-8(u74) I*; *pha-1(tm3671) III*; *sym-3(mn618) X*; *mnEx169*; *fdEx201 (PBX(pha-1(+)*; *sur-5::RFP)*)], WY965 [*lin-35(n745) I*; *sym-3 (mn618) X*], WY964 [*lin-35(n745) I*; *sym-4(mn619) X*], GE24 [*pha-1(e2123ts) III*], WY849 [*pha-1 (tm3671) III*; *fdEx183 (pBX*; *sur-5::GFP)*], CHB11 [*fbn-1(ns67) III*; *oyIs44 [odr-1::RFP] V*], CHB31 [*fbn-1(ns283) unc-32(e189) III*; *kyIs136 (str-2pro::GFP) X*], WY1034 [*fbn-1(tm290) III*; *fdEx249 (sur-5::GFP*; *fbn-1(+)-fosmid wrm0635cH08)*], WY1048 [*fbn-1(ns67) III*; *oyIs44 V*; *sym-3(mn618) X*; *mnEx169*], WY1049 [*fbn-1(ns67) III*; *oyIs44 V*; *sym-4(mn619)*; *fdEx225 (sym-4(+)*; *sur-5::GFP)*], WY1056 [*mec-8(u74) I*; *fbn-1(ns67) III*; *oyIs44*; *fdEx249*], WY1057 [*fbn-1(tm290) III*; *sym-3(mn618) X*; *fdEx249*], WY1058 [*fbn-1(tm290) III*; *oyIs44 V*; *sym-3(mn618) X*; *fdEx249*], WY1068 [*mec-8(u74) I*; *fbn-1(tm290) III*; *fdEx249*], CB4121 [*sqt-3(e2117) V*], *mec-8(u74) I*; *sqt-3(e2117) V*], ARF256 [*aaaEx32 (Pfbn-1::gfp-pest + Pttx-3::GFP)*], ARF262 [*aaaEx33 (fbn-1Δ49-2418::mCherry + P_{myo-2}::GFP)*], WY1082 [*fbn-1(ns67)*; *aaaEx32*], GN517 [*pgEx116 (unc-70-TSmod*; *myo3::mCherry)*], GN519 [*pgEx131 (unc-70(5aa) punc-122::RFP)*], GN518 [*pgEx126 (unc-70(TRAF)*; *punc-122::RFP)*], GN600 [*pgIs22 (unc-70 (N-TSMod)*), *oxIs95 (myo2::gfp*; *pdi-2::unc-70)V*], WY1047 [*pha-1(tm3671) III*; *fdEx182*; *pgEx116*], GN486 [*unc-70(s1502)V*; *oxIs95 IV*; *pgEx126*], GN491 [*unc-70(s1502)V*; *oxIs95IV*; *pgEx131*], GN601 [*unc-70(s1502) V*; *oxIs95 IV*; *pgIs22*], NR222 [*rde-1(ne219) V*; *kzIs9 (pKK1260(lin-26p::nls::GFP))*; *pKK1253(lin-26p::rde-1)*; *pRF6(rol-6(su1006))*], WY1033 [*mec-8(u74) X*; *rde-1(ne219) V*; *kzIs9*].

## Tension sensor studies

### Generation of transgenic animals

A detailed description of the molecular cloning of the *unc-70* cDNA and TSMod derivatives is found in *Krieg et al., 2014*. Transgenesis was performed by microinjection following standard procedures. We also assayed the ability of the UNC-70(TSMod) as well as the low FRET, high FRET and no-force transgenes to rescue the locomotion defect of *unc-70(s1502)* mutants. To do so, we placed transgenic animals onto fresh agar plates and recorded short movies (<1 min) and compared the movement (curvature matrices) of transgenic animals to the parental *unc-70(s1502);oxIs95* mutants (*Hammarlund et al., 2007*) and to wild-type animals. All constructs [UNC-70(TSMod) *pgEx116*, UNC-70(TRAF) *pgEx126*, UNC-70(5aa) *pgEx131* and UNC-70(N-TSMod) *pgIs22*] were capable of restoring locomotion to paralyzed *unc-70(s1502);oxIs95* adults (data not shown).

### Imaging acquisition

Förster resonance energy transfer (FRET) images were acquired and processed as described in detail in *Krieg et al., 2014*. In short, three images for each focal plane and time point were taken at $512 \times 512$ pixels and with a 400-Hz acquisition rate using a Leica SP5 confocal microscope. The three images were: (1) donor emission after direct donor excitation, (2) acceptor emission after direct excitation of the acceptor and (3) acceptor emission after excitation of the donor, representing the raw uncorrected image. In total, a z-stack of the whole embryo was taken, and frames encompassing the buccal cavity were analyzed (3–5 frames on average). Before processing, all images were binned (downsampled and averaged) once to increase the signal-to-noise ratio and imported into IgorPro (WaveMetrics, Oregon) for further processing. A pre-calibration of the microscope using a solution of 0.01% fluorescein showed that detector gains are linear within the laser power range used for these studies.

### Image processing

The raw FRET image was background subtracted and corrected for bleed-through using pre-determined values for each bleed-through factor (*Krieg et al., 2014*) according to $cF(i, j) = I_F(i, j) - \delta \cdot I_D(i, j) - \alpha \cdot I_A(i, j)$ in which $\alpha$ and $\delta$ are the measured bleed-through factors for the acceptor and the donor channel, respectively (*Krieg et al., 2014*), and i, j are the pixel coordinates. The final FRET index image was calculated according to

$$F = \frac{cF \cdot Q_D \cdot \varphi_{D/\varphi_A}}{qD + \left(cF \cdot Q_D \cdot \varphi_{D/\varphi_A}\right)}$$

on a pixel-by-pixel basis (*Chen and Periasamy, 2006*; *Krieg et al., 2014*)

in which $cF$ is the bleed-through corrected FRET channel intensity, $Q_d$ is the quantum yield of the donor fluorophore and was empirically determined (*Day et al., 2008*), and $qD$ is the quenched

donor intensity. The ratio $\varphi_D/\varphi_A$ is the collection efficiency and has been determined empirically (**Krieg et al., 2014**). A [3 × 3] median filter was applied to remove high-frequency noise in the FRET map.

## Final quantification and ROI selection

The FRET index image was quantified by manually selecting a box (region of interest [ROI], **Figure 3**) over the putative pharynx attachment site to the epidermis (sensory depression region, SDR). More specifically, the SDR ROIs were identified as the area of the highest intensity and curvature around the sensory depression of the epidermis. In *pha-1* mutants, the sensory depression was identified as the region directly anterior to the (non-attached) primordial pharynx. A second ROI (non-SDR) was then chosen that encompassed the remainder of the embryo (excluding the SDR) and, therefore, was larger than the ROI of the SDR. The rational was to create an 'internal' reference value, to which the SDR can be compared, which is independent of differences in sensor expression level, variations in bleed-through, bleaching, and downstream image processing procedures, as every pixel in an image has the exact same history. Because the precise size of the ROIs varied between embryos, the number of pixels within individual SDR ROIs and non-SDR ROIs was also variable. A pairwise comparison between SDR and non-SDRs within individual embryos was carried out to assess variations of FRET values independent of different expression levels among different embryos.

To evaluate the quality of our measurements and robustness against uncontrolled changes in intensity due to stochastic shot noise, we calculated the uncertainty of the FRET values derived from UNC-70(N-TSMod) embryos (since they do not show differences in FRET due to tension) in each pixel by error propagation (**Berney and Danuser, 2003**). Pixels with low intensity show high uncertainty. To minimize the impact of this source of error, we limited our analysis to those pixels within each ROI whose intensity exceeded a threshold. This strategy also selects domains in which the sensor is concentrated (i.e., cell cortices) and excludes those that lack sensor molecules (i.e., cytoplasm). No significant correlation of the FRET index with the expression level (acceptor counts) was observed, indicating that intermolecular FRET had a negligible contribution to the final FRET index image (data not shown; **King et al., 2014**).

## Statistical evaluation

FRET index values from different planes encompassing the buccal cavity within a given embryo were pooled, and the average was treated as an experimental value. Statistical significance was assessed using a two-tailed t-test to compare intra-embryonic ROIs and ROIs between embryos. Data were normally distributed as determined using the Jarque–Bera test for normality and exhibited equal variance as judged by Levene's test. Comparison of FRET values was carried out using a Student's T test and paired T-tests (where applicable; **Supplementary file 2**) and the Mann–Whitney U-test (data not shown).

## Tiling array studies

Wild-type *C. elegans* (N2) and *mec-8*(*u314*) animals were grown at 20°C on high peptone plates until gravid. Embryos were extracted with a solution that contained 1 M NaOH and 30% bleach, in water. Total RNA from embryos was extracted using TriReagent (Sigma) and cleaned using RNeasy columns (Qiagen) according to the manufacturer's protocol. Purified RNA was then treated with 10 U DNase I (Roche) for 30 min in 100 μl 1× One-Phor-All buffer (Amersham). The RNA was then re-purified with RNeasy columns (QIAGEN) and 1 μl random hexamers (3 μg/μl) was added to 15 μg purified total RNA together with reverse transcriptase. The (ds)cDNA was then purified using QIAGEN PCT purification columns, and 17 μg of (ds)cDNA was digested and labeled using standard Affymetrix methods The hybridization cocktail was injected into an Affymetrix GeneChip *C. elegans* Tiling 1.0R Array. Hybridized microarrays were washed and scanned according to chapter 5 of the GeneChip Whole Transcript (WT) Double Stranded Target Assay Manual (https://www.affymetrix.com/support/downloads/manuals/wt_dble_strand_targe-t_assay_manual.pdf). For reverse transcriptase PCR studies of select MEC-8 targets identified by the tiling array (**Figure 4—figure supplement 1**), total RNA used for the tilling array experiments was reverse transcribed using oligo-dT primers and amplified using specific primers for each gene region (**Supplementary file 3**) for 30 cycles.

## PCR analysis of *fbn-1*

Total RNA from N2 and *mec-8(u74)* embryos was isolated using Trizol and purified on RNeasy minicolumns (Qiagen). cDNA was prepared from 1 μg RNA using a SuperScript II first-strand synthesis system (Invitrogen) and analyzed by PCR (30 and 35 cycles) using the following primers: 5′-CAACAGAGTCATCCGAAGCT-3′ and 5′-TGCAGTTGTGGTGGTGGTAGGT-3′ (which anneal to exon 16 and exon 17 of *fbn-1*, respectively), 5′-GACAGGAAAAACCAACTACTAAA-3′ and 5′-TGTGACTGTGGAGCAAAGAGATG-3′ (which anneal to exon 14 and exon 19 of *fbn-1*, respectively) and 5′-TGTCTTCCAGGATTTACTGGAG-3′ and 5′-TACATACTGCGTTCGGGTG-3′ (which anneal to exon 19 and exon 20 of *fbn-1*, respectively).

## RNAi

RNAi-feeding was done with strains from the Geneservice Library, using the standard feeding protocol (*Ahringer, 2006*). Control RNAi-feeding assays were carried out using a bacterial strain carrying the RNAi vector pDF129.36, which produces an ~200-bp dsRNA that is not homologous to any *C. elegans* gene (*Timmons et al., 2001*). RNAi hypersensitive mutations used included *lin-35* (*n745*) (*Wang et al., 2005*; *Lehner et al., 2006*) and *rrf-3(pk1426)* (*Simmer et al., 2002*). For *let-502* (*RNAi*) of *mec-8*; *fbn-1* mutants, dsRNA targeting *let-502* exon 4 was injected into P0s at a concentration of ~500 ng/μl and F1s laid between 24–48 hr post injection were scored.

## Mosaic analysis

Mosaic analysis was carried out using strains WY1059, *fbn-1(tm290)*; *sym-4(mn619)*; *fdEx249*[*fbn-1*(+); *sur-5::GFP*], and WY1068, *mec-8(u74)*; *fbn-1(tm290)*; *fdEx249*, following established protocols (*Yochem et al., 2000*; *Yochem, 2006*).

## Expression analysis and DNA constructs

To generate P$_{fbn-1}$::GFP-PEST, nucleotides 7621743–7625323 of Chromosome III were amplified from N2 and spliced to the *gfp-pest* cassette from pAF207 (*Frand et al., 2005*) using PCR methods. We note that strains expressing P$_{fbn-1}$::GFP-PEST also contain a P$_{ttx-3}$::GFP marker, which is expressed in AIY neurons but not in epidermal cells (*Hobert et al., 1997*). To generate the mini-*fbn-1*::mCherry (*fbn-1Δ49-2418*::mCherry) fusion gene, nucleotides 7621652–7626214 of Chromosome III, which contain the presumptive *fbn-1* promoter and first 48 codons, were amplified from N2 DNA and cloned into pUC19. Then nucleotides 7638794–7641181 of Chromosome III, which contain the last 362 codons and native 3′ UTR of *fbn-1*, were amplified from N2 and inserted downstream of the former fragment, producing plasmid pVM61. The mCherry cassette from KP1272 was then inserted in-frame between the two genomic fragments using an engineered NotI site. Injection of DNA constructs or PCR products to generate extrachromosomal arrays was carried out using standard procedures (*Mello and Fire, 1995*).

A rescuing *fbn-1* cDNA sequence was PCR amplified as three overlapping fragments (atgtctac... gaaaattg, 2049 bp; ggaaaagt...gtacctgc, 3581 bp; gtatggct...gattctag, 2758 bp) from a cDNA library (gift of Carl Procko) and the plasmid clone yk670d9 (gift of Yuji Kohara), which were then assembled into a single 8065-bp cDNA sequence using internal PstI and SalI sites (at position 1985 bp and 5431 bp, respectively). A 4406-bp *fbn-1* promoter sequence capable of driving embryonic GFP expression was isolated (tcgaggag...ttgcagga) and assembled with the *fbn-1* cDNA as a SbfI-AgeI promoter fragment and an AgeI-NheI cDNA fragment in a modified pPD95.69 vector bearing an NheI-SpeI unc-54 3′ UTR fragment, to create the plasmid pMH281.

## Microscopy

With the exception of FRET studies, micrographs were taken with a Nikon Eclipse microscope, using a 100× objective. Percent Pin was calculated by counting 1.5-fold or older embryos or L1-stage larvae. Fluorescent confocal images were acquired using a 100× objective on an Olympus IX-71 inverted microscope. Image acquisition and microscope control were carried out with Metamorph software (Molecular Devices).

Keyhole/sensory depression depth was quantified using Openlab software. Keyhole depth for *mec-8*; *sym-3* embryos grown on *let-502* RNAi plates was quantified every 15 min starting at the late comma stage through the threefold stage. *mec-8*; *sqt-3* embryo keyhole depth was measured by growing embryos at 25°C starting at the twofold stage and quantified every 60 min for 5 hr.

## Trypsin treatment of embryos

Embryos were obtained by bleaching N2 adults, using standard methods. Embryos were permeabilized by treatment with 2 mg/ml chitinase for 5–10 min at room temperature (*Bianchi and Driscoll, 2006*). Permeabilized mixed-stage embryos were treated with trypsin (Sigma Aldrich) at a concentration of 5 µg/ml for 15 min at room temperature followed by trypsin inhibitor (Sigma Aldrich), which was added to a concentration of 50 µg/ml and incubated for 2 min (*Priess and Hirsh, 1986*). Embryos were rinsed with M9 and examined immediately using DIC microscopy.

## Acknowledgements

Some strains were provided by the *Caenorhabditis* Genetics Center (CGC), which is funded by the US National Institutes of Health (NIH) Office of Research Infrastructure Programs (P40 OD010440). We also thank the National BioResource Project of Japan and Eric Jorgensen for strains. FRET imaging was conducted in the Cell Sciences Imaging Facility at Stanford, which is supported by award S10RR02557401 from the National Center for Research Resources. MBG and MK were supported by NIH grants NS047715, EB006745 and 1K99NS089942-01 as well as by a Human Frontier Science Program Long-Term Fellowship (MK), respectively. AC and MC were supported by NIH grant GM30997. AC was also supported by FONDECYT grant 1131038. AF and VM were supported by the NMF and ACS (RSG-12-149-01-DCC). SS and MH were supported in part by NIH grants NS073121and NS064273. MH was also supported in part by NIH grant GM108754. Support at the University of Wyoming was from the NIH (R01 grant GM066868 to DSF and INBRE grant P20 GM103432). We thank Amy Fluet for editing, Dan Starr for valuable scientific input, and Ronald Tepper for help with the microarray data analysis.

## Additional information

### Funding

| Funder | Grant reference | Author |
| --- | --- | --- |
| Human Frontier Science Program (HFSP) | Long-Term Fellowship | Michael Krieg |
| American Cancer Society | RSG-12-149-01-DCC | Vijaykumar Meli, Alison Frand |
| National Marfan Foundation (NMF) | RSG-12-149-01-DCC | Vijaykumar Meli, Alison Frand |
| National Institutes of Health (NIH) | NS047715 | Melissa Kelley, Miriam B Goodman |
| National Institutes of Health (NIH) | EB006745 | Melissa Kelley, Miriam B Goodman |
| National Institutes of Health (NIH) | 1K99NS089942-01 | Melissa Kelley, Miriam B Goodman |
| National Institutes of Health (NIH) | GM30997 | Andrea Calixto, Martin Chalfie |
| Fondo Nacional de Desarrollo Científico y Tecnológico | 1131038 | Andrea Calixto |
| National Institutes of Health (NIH) | NS073121 | Maxwell G Heiman, Shai Shaham |
| National Institutes of Health (NIH) | NS064273 | Maxwell G Heiman, Shai Shaham |
| National Institutes of Health (NIH) | GM108754 | Maxwell G Heiman |
| National Institutes of Health (NIH) | GM066868 | David S Fay |
| National Institutes of Health (NIH) | P20 GM103432 | David S Fay |

The funders had no role in study design, data collection and interpretation, or the decision to submit the work for publication.

## Author contributions
MK, DSF, Conception and design, Acquisition of data, Analysis and interpretation of data, Drafting or revising the article, Contributed unpublished essential data or reagents; JY, MK, AC, MGH, AK, Conception and design, Acquisition of data, Analysis and interpretation of data, Drafting or revising the article; VM, Conception and design, Acquisition of data, Analysis and interpretation of data; MC, MBG, SS, AF, Conception and design, Analysis and interpretation of data, Drafting or revising the article

## Author ORCIDs
Martin Chalfie, http://orcid.org/0000-0002-9079-7046

# Additional files

## Supplementary files
• Supplementary file 1. Percentage of Pin Animals.

• Supplementary file 2. Statistical analysis of FRET data.

• Supplementary file 3. *mec-8* tiling array data. (**A**) Gene Regions. Gene regions determined to be differentially expressed in mec-8 animals as compared with wild type. Genes for which all exons were up or down regulated are in bold. Region indicates the specific exons or introns of a transcript that were up or down regulated in mec-8 animals. Note that exon or intron numbering is specific to individual isoforms (as indicated) and may differ between isoforms. For example, the region encompassing intron 14 in fbn-1b, d, e corresponds to introns 14–15 in fbn-1a, k and the region encompassing intron 15 in fbn-1e corresponds to introns 16–18 in fbn-1a, k. Start and End indicate the flanking genomic positions of the differentially expressed exons or introns. Difference is the length in nucleotides of the exons or introns that were differentially expressed. Probes indicate the actual number of 25-nt probes that hybridized the indicated region. D/25 is the theoretical number of 25- nucleotide (nt) probes that cover the length of the genomic fragment. N2 avg and mec- 8 avg are the averaged intensities from three replicates. N2 SD and mec-8 SD are the standard deviation of the three replicates. Ratio is the average intensity of mec-8 divided by the average intensity of N2. SD ratio is the standard deviation of the mec- 8/N2 ratio of intensity. AS (alternatively spliced) indicates whether the gene is alternatively spliced. Gene ratios are color coded according to groups: orange, introns up; green, exons down; red, exons up. fbn-1 and mec-8 gene names are highlighted in red. In some cases different regions from the same gene are separated based on ratio rankings. (**B**) Transcripts. Transcripts differentially expressed in mec-8 animals compared to wild type. Average ratio is the average of the ratios of each gene region of the transcript. (**C**) Gene lists. Alphabetical list of all differentially expressed genes including category subdivisions. (**D**)Alternatively (Alt.) Spliced Genes. Alphabetical list of all differentially expressed genes including category divisions and information regarding alternative splicing. (**E**) PCR Primers. Sequences of primers (5′–3′) used to amplify candidate gene regions. Expected sizes for candidates that are differentially expressed in mec-8 mutants are indicated. Orange indicates introns upregulated in mec-8; red, exons upregulated in mec-8; blue, exons downregulated in mec-8. For genes that contain intronic regions that are upregulated in mec-8, sizes with and without, respectively, intronic sequences are shown.

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
