## [Decision Letter]

Thank you for sending your work entitled “FBN-1 is required for resistance of the epidermis to mechanical deformation during *C. elegans* embryonic development” for consideration at *eLife*. Your article has been favorably evaluated by Janet Rossant (Senior editor) and three reviewers, one of whom is a member of our Board of Reviewing Editors.

The Reviewing editor and the other reviewers discussed their comments before we reached this decision, and the Reviewing editor has assembled the following comments to help you prepare a revised submission.

Morphogenesis involves the generation and correct localization of forces. Although many molecules known to be required for morphogenesis are known, our understanding of how they function and their relationship with force is limited. Using genetic, phenotypic analyses, and a FRET-based tension sensor, the authors identify and characterize a mechanical force that occurs during development of the *C. elegans* mouth and show that FBN-1 plays a role in resisting this inward pulling force. They also identify genes required for proper *fbn-1* expression. This is an interesting overall study that advances our understanding of the processes and factors controlling morphogenetic movements and should be of wide interest.

Points that need to be addressed:

1) The experiments with the three control FRET sensors are important validations of the FRET sensor. Details of their construction should be included in the Methods. Also, do the control UNC-70 sensors UNC-70(TRAF) and UNC-70(5aa) rescue *unc-70* mutants as the FRET sensor does? This is an important validation of the control sensors.

2) How regions outside the sensory depression were chosen for measurement in the FRET experiments is not clear from the Methods. The authors said that they selected an ROI over the putative pharynx attachment site and another ROI “over the rest of the embryo.” The Results section says that pixels with similar expression levels were chosen. Were the ROIs the same size? Where in the embryo were the chosen non-sensory depression ROIs? How were they chosen?

3) There are some concerns regarding interpretation of the force sensor data. The authors record molecular tension, which cannot necessarily be translated into tissue tension given that the molecular machinery transmitting forces within and between tissues are not entirely known. In the absence of experiments testing mechanical resistance of the embryonic sheath, the authors should more clearly describe the limitations of the force sensor data for supporting their mechanical model.

---

## [Author Response]

*1). The experiments with the three control FRET sensors are important validations of the FRET sensor. Details of their construction should be included in the Methods. Also, do the control UNC-70 sensors UNC-70(TRAF) and UNC-70(5aa) rescue* unc-70 *mutants as the FRET sensor does? This is an important validation of the control sensors*.

Details on the construction of the FRET sensors are found in [69]. We have made this point clear in the Methods section. We have also now tested all three FRET sensor controls (UNC-70(TRAF), UNC-70(5aa), and UNC-70(N-TSMod) in the *unc-70(s1502)* background. We find that all three sensor controls, like UNC-70(TSMod), rescue the paralysis of *unc-70* mutants, further validating the use of these controls (subsections “A FRET-based tension sensor reveals mechanical forces operating during embryogenesis” and “Tension sensor studies”).

*2) How regions outside the sensory depression were chosen for measurement in the FRET experiments is not clear from the Methods. The authors said that they selected an ROI over the putative pharynx attachment site and another ROI “over the rest of the embryo.” The Results section says that pixels with similar expression levels were chosen*. *Were the ROIs the same size? Where in the embryo were the chosen non-sensory depression ROIs? How were they chosen?*

We have now clarified issues regarding ROI selection of the sensory depression and non-sensory depression regions (in the Results and in the Materials and methods sections).

*3) There are some concerns regarding interpretation of the force sensor data. The authors record molecular tension, which cannot necessarily be translated into tissue tension given that the molecular machinery transmitting forces within and between tissues are not entirely known. In the absence of experiments testing mechanical resistance of the embryonic sheath, the authors should more clearly describe the limitations of the force sensor data for supporting their mechanical model*.

This is a good point and we have clarified issues concerning the interpretation of the force sensor data (please see the subsection “A FRET-based tension sensor reveals mechanical forces operating during embryogenesis” and the Discussion).

We have also complied with the request of *eLife* editor, Peter Rogers, to add a descriptor for FBN-1 in the title. The new title is: FBN-1, a fibrillin-related protein, is required for resistance of the epidermis to mechanical deformation during *C. elegans* embryogenesis.